# Burial-induced oxygen-isotope re-equilibration of fossil foraminifera explains ocean paleotemperature paradoxes

S. Bernard[1], D. Daval[2], P. Ackerer[2], S. Pont[1] & A. Meibom[3,4]

Oxygen-isotope compositions of fossilised planktonic and benthic foraminifera tests are used as proxies for surface- and deep-ocean paleotemperatures, providing a continuous benthic record for the past 115 Ma. However, visually imperceptible processes can alter these proxies during sediment burial. Here, we investigate the diffusion-controlled re-equilibration process with experiments exposing foraminifera tests to elevated pressures and temperatures in isotopically heavy artificial seawater ($H_2^{18}O$), followed by scanning electron microscopy and quantitative NanoSIMS imaging: oxygen-isotope compositions changed heterogeneously at submicrometer length scales without any observable modifications of the test ultrastructures. In parallel, numerical modelling of diffusion during burial shows that oxygen-isotope re-equilibration of fossil foraminifera tests can cause significant overestimations of ocean paleotemperatures on a time scale of $10^7$ years under natural conditions. Our results suggest that the late Cretaceous and Paleogene deep-ocean and high-latitude surface-ocean temperatures were significantly lower than is generally accepted, thereby explaining the paradox of the low equator-to-pole surface-ocean thermal gradient inferred for these periods.

[1] IMPMC, Sorbonne Universités, CNRS UMR 7590, MNHN, UPMC, IRD UMR 206, 61 Rue Buffon, 75005 Paris, France. [2] LHyGeS, CNRS UMR 7517, Université de Strasbourg/EOST, 1 Rue Blessig, 67084 Strasbourg, France. [3] Laboratory for Biological Geochemistry, School of Architecture, Civil and Environmental Engineering, École Polytechnique Fédérale de Lausanne (EPFL), 1015 Lausanne, Switzerland. [4] Center for Advanced Surface Analysis, Institute of Earth Sciences, University of Lausanne, 1015 Lausanne, Switzerland. Correspondence and requests for materials should be addressed to S.B. (email: sbernard@mnhn.fr)

Precise knowledge of the past ocean temperature is essential for understanding hydrosphere evolution and for placing anthropogenic global climate change in a geologic perspective. Since the early 1950s, the O isotope compositions of fossil foraminifera tests have been an important tool in paleoclimate research[1,2]. Field and laboratory studies have demonstrated that the $^{18}O/^{16}O$ ratio of living foraminifera calcite tests is a function of both the temperature and the O isotope composition of the seawater, the latter of which varies with global ice volume, pH and salinity[3–5]. The most widely accepted interpretation of the continuous benthic foraminifera O isotope record for the past 115 Myr is that the Cretaceous deep ocean was very warm and continuously cooled by ~15 °C during the late Cretaceous and the Paleogene[6–9].

The corresponding planktonic O isotope record was initially interpreted to indicate relatively warm high-latitude sea-surface temperatures and relatively cold tropical sea-surface temperatures during the late Cretaceous and the Paleogene[10], giving rise to the cool tropics paradox[11]. This paradox was related to secondary calcite precipitation that distorts paleotemperature reconstructions[12–16]. In fact, tests showing no visible alterations, also known as glassy foraminifera, yielded significantly higher temperatures for the Cretaceous and Paleogene low-latitude surface oceans[17,18]. Still, the currently accepted interpretation of the planktonic and benthic foraminifera records is that both the vertical thermal gradient in the tropical ocean and the equator-to-pole surface-ocean temperature gradient were much less steep during these periods than those in the present ocean[6–10,17,18]. However, such thermal gradients cannot be reconciled with the most recent climate and ocean circulation models[19,20].

The O isotope composition of fossil foraminifera tests can be modified during burial without any visible structural changes. Despite the early warning of Urey et al.[21] and recent evidence that carbonates can undergo isotope re-equilibration at low temperatures[22], the potential bias of paleotemperature estimates resulting from such visually imperceptible processes has never been quantified. Isotope re-equilibration of foraminifera tests with sediment pore water likely occurs over millions of years during sediment burial at relatively low temperatures (~20–30 °C) and pressures (200–500 bars). However, such a re-equilibration cannot be visualised in natural specimens, even when using high-resolution analytical techniques, such as the NanoSIMS ion microprobe. In fact, although the NanoSIMS can quantify isotope compositions at a lateral resolution of ~100 nm[23], it does not have the required analytical precision to reveal the resulting permil-level variations.

Here, we conducted experiments with water strongly enriched in $^{18}O$ at high temperatures to visualise the burial-induced isotope re-equilibration of foraminifera tests using quantitative NanoSIMS imaging. The results demonstrate that the O isotope compositions of foraminifera tests can change without any observable modifications of their ultrastructures. Numerical simulations suggest that the late Cretaceous and Paleogene deep-ocean and high-latitude surface-ocean temperatures were significantly lower than is generally accepted.

## Results

**Isotope re-equilibration experiments.** Sealed gold capsules, each containing 160 μg of cleaned foraminifera tests (i.e. ~12 specimens of *Globigerina bulloides* (bulk $\delta^{18}O = 1.35‰$ Vienna Pee Dee Belemnite (VPDB) ($\pm 0.05‰$; $2\sigma$)) from modern sediments of the Lion Gulf, France) and 100 μL of an artificial $H_2^{18}O$ seawater solution with 0.55 mol L$^{-1}$ of NaCl and 0.003 mol L$^{-1}$ of NaHCO$_3$, were submitted to a temperature of 300 °C under a confining pressure of 200 bars for 3 months. Control experiments were performed under identical conditions with chemically similar but isotopically normal artificial seawater. The foraminifera tests were ultrasonically cleaned in pure ethanol before these experiments to remove the clays attached to their surfaces and were cleaned after the experiments to remove as much of the $^{18}O$-pure artificial seawater adhering to their surfaces as possible. The fine-scale morphology of these tests was documented before and after the experiments with scanning electron microscopy (SEM). Finally, the tests were embedded in epoxy resin and polished for quantitative NanoSIMS isotope imaging.

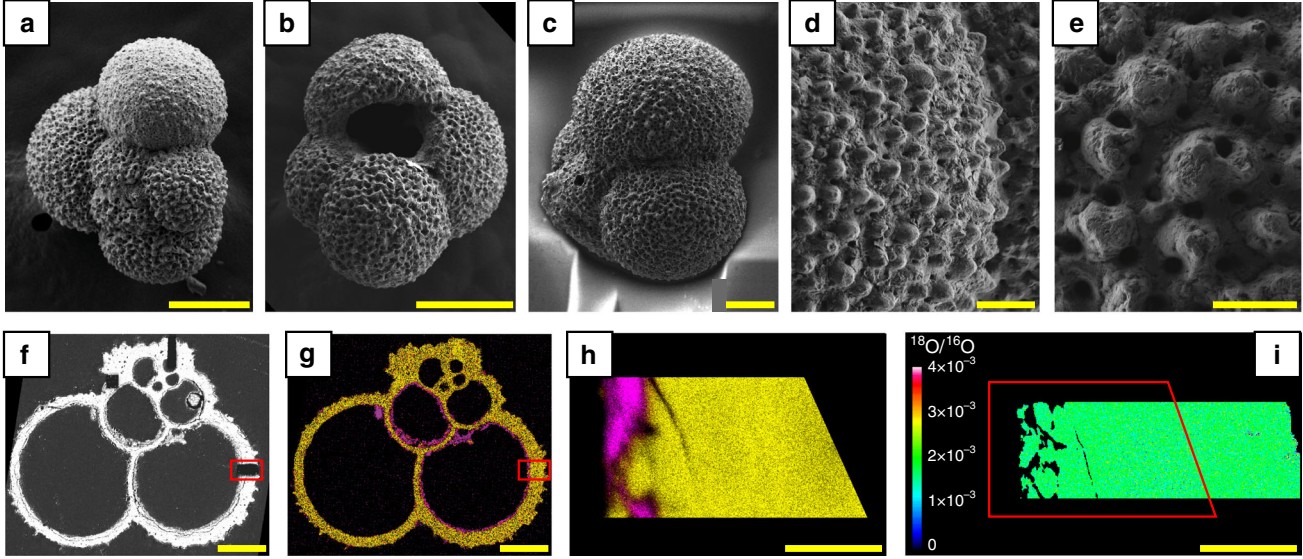

**Fig. 1** Foraminifera tests prior to isotope re-equilibration experiments. **a–e** SEM images showing the overall morphologies and ultrastructures of the foraminifera tests (*G. bulloides*) after their ultrasonic cleaning in pure ethanol. **f** Backscattered SEM image of a polished section of a test embedded in epoxy. **g, h** Energy dispersive X-ray spectroscopy (EDXS) maps showing the spatial distributions of carbonates (Ca appears in yellow) and clays (Si appears in pink). **i** NanoSIMS map displaying the isotopically homogeneous nature of the carbonates and clays ($^{18}O/^{16}O = 0.002$). The red rectangles in **f**, **g** indicate the locations of **i**. The red rectangle in **i** indicates the location of **h**. Scale bars are 100 μm (**a**, **b**, **f**, **g**), 50 μm (**c**), 20 μm (**d**, **i**) and 10 μm (**e**, **h**)

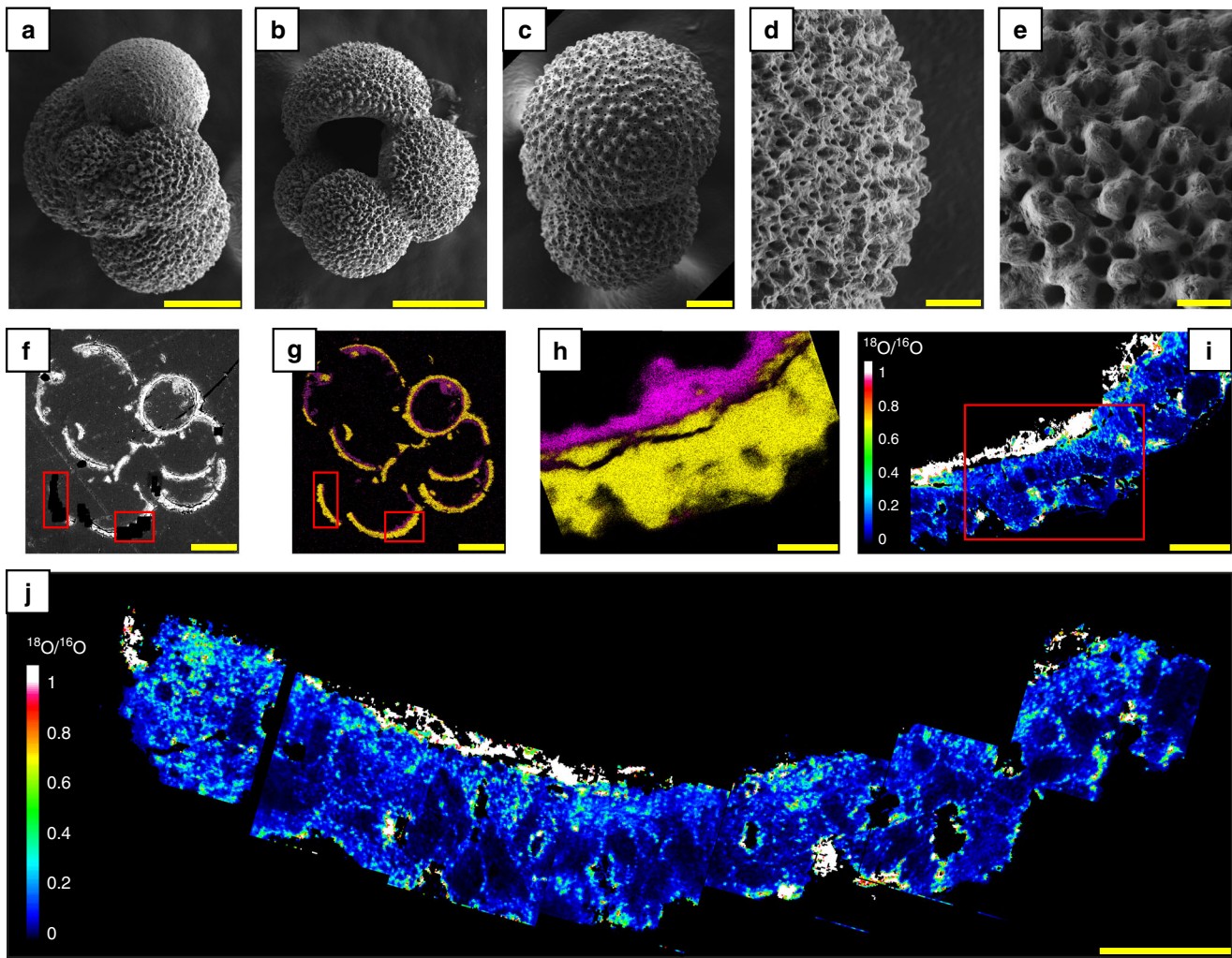

**Fig. 2** Foraminifera tests after isotope re-equilibration experiments. **a–e** SEM images showing the overall morphologies and ultrastructures of the foraminifera tests (*G. bulloides*) after their ultrasonic cleaning in pure ethanol. **f** Backscattered SEM image of a polished section of a test embedded in epoxy. **g, h** EDXS maps showing the spatial distributions of carbonates (Ca appears in yellow) and clays (Si appears in pink). **i, j** NanoSIMS maps showing the $^{18}O/$$^{16}O$ ratio distributions in carbonates and clays. Carbonates exhibit a $^{18}O/^{16}O$ ratio ranging from 0.1 to 0.8, while clays exhibit a $^{18}O/^{16}O$ ratio ranging from 0.9 to greater than 1.0. Note that the isotope exchange occurred very heterogeneously, leading to areas that are more or less enriched in $^{18}O$. The red rectangles in **f, g** indicate the locations of **i, j**. The red rectangle in **i** indicates the location of **h**. Scale bars are 100 μm (**a, b, f, g**), 50 μm (**c**), 20 μm (**d, i**) and 10 μm (**e, h, j**)

The morphology of the foraminifera tests that were submitted to these experiments did not discernibly differ from the starting materials, even at submicrometer length scales (Figs. 1 and 2). In contrast, the NanoSIMS imaging revealed that the average $^{18}O/^{16}O$ ratio of the tests immersed within the $^{18}O$-pure artificial seawater, initially 0.002 (Fig. 1), reached up to 0.15 during the experiments (Fig. 2), which is equivalent to the replacement of ~15 vol% of the initial biogenic calcite by pure $CaC^{18}O^{18}O^{18}O$. Note that areas extremely enriched in $^{18}O$ were observed (Fig. 2), showing that the isotope exchange did not occur homogeneously within the biogenic calcite matrix. These results unambiguously demonstrate a process in which the foraminifera tests undergo modifications to their bulk O isotope composition without any appreciable modification of their fine scale morphology, raising serious questions about the reliability of fossil foraminifera tests (including the so-called glassy specimens) as proxies of seawater temperatures in the geological past.

**Isotope re-equilibration processes**. Mechanistically, the observed O isotope exchange can occur either through coupled dissolution and reprecipitation at mineral–fluid interfaces (replacement that leads to the formation of secondary calcites while retaining the original shape of the primary calcite structures)[24,25], or through (much slower) solid-state grain boundary and volume diffusion (movement and transport of oxygen atoms within the biogenic calcite matrix without dissolution)[26–28]. Coupled dissolution and reprecipitation at mineral–fluid interfaces requires the aqueous phase and the dissolving solids to remain in contact to propagate a replacement front[24,25]. This replacement process is a volume deficit reaction that generates porous secondary products[24,25]. Because there is no volume change in the replacement of $^{18}O$-poor calcite by $^{18}O$-rich calcite, this process is not considered here. In the following, we thus investigate the O isotope re-equilibration of foraminifera tests during sediment burial under the conservative assumption that this re-equilibration occurs exclusively through solid-state grain boundary and volume

diffusion, thereby exploring the slowest process that can alter the original foraminifera isotope signal.

Once incorporated into sediments, fossil foraminifera tests experience increasing temperatures with increasing sediment burial depths. Because O isotope fractionation between calcite and water decreases with increasing temperature (see 'Methods'), the burial-induced increase of the sediment temperature establishes isotopic disequilibrium between the tests and the surrounding pore water. This disequilibrium constitutes the driving force for diffusion, which works continuously towards re-establishing isotopic equilibrium between the pore water and tests.

Electron and atomic force microscopy of the foraminifera tests show that they consist of 50–250 nm calcite domains[29,30] embedded in an organic matrix that is highly prone to degradation at shallow burial depths[31]. This, in combination with the formation of cracks of various length scales, allows most of the submicrometric calcite domains of fossil foraminifera tests to be directly in contact with pore water. Subsequent solid-state grain boundary and volume diffusion occurs along the boundaries and into individual nanoscale calcite domains. In natural settings, at temperatures characterising a typical sedimentary stack, solid-state grain boundary and volume diffusion in calcite is a slow process that occurs over very small distances (<tens of nm). However, given the small sizes of the calcite domains[29,30], diffusion can significantly impact the bulk O isotope composition of the foraminifera tests over geologic timescales. Because full re-equilibration cannot be achieved in natural settings on relevant timescales, fossil foraminifera tests remain isotopically heterogeneous at the nanoscale, as is the case in the present experiments (Fig. 2).

**Isotope re-equilibration impact**. Using numerical modelling, we quantified changes in fossil foraminifera test O isotope composition in response to burial-induced O isotope re-equilibration through solid-state diffusion assuming a constant chemical composition of the pore water. Here, the sediment pore water was assumed to have the same O isotope composition as the seawater in which foraminifera formed[32], i.e. a $\delta^{18}O$ of −1‰ Vienna Standard Mean Ocean Water (VSMOW), consistent with the absence of polar ice caps[2]. Realistic values were chosen for the burial rate, the geothermal gradient and the activation energy. The burial rate (i.e. the sedimentation rate) was adopted from a compilation of the representative DSDP, ODP and IODP sites (http://deepseadrilling.org/; Fig. 3a). Geothermal gradients from 40 to 60 °C km⁻¹, a range consistent with reported observations[31], were tested. Note that in these scenarios, foraminifera are never exposed to temperatures exceeding 25–30 °C (Fig. 3a). The activation energy for oxygen diffusion was varied between 85 and 95 kJ mol⁻¹, consistent with values proposed in the literature[26,27] (see 'Methods').

Figure 3 shows the results of the simulations that assume benthic foraminifera tests that formed in equilibrium with a

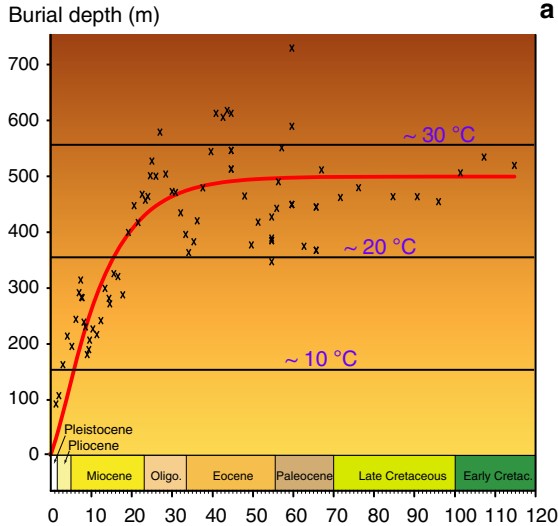

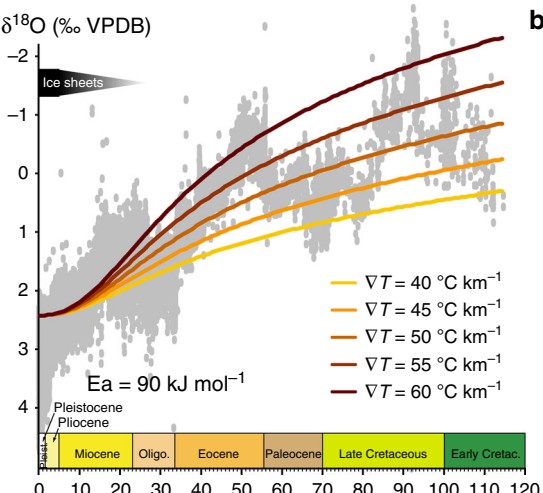

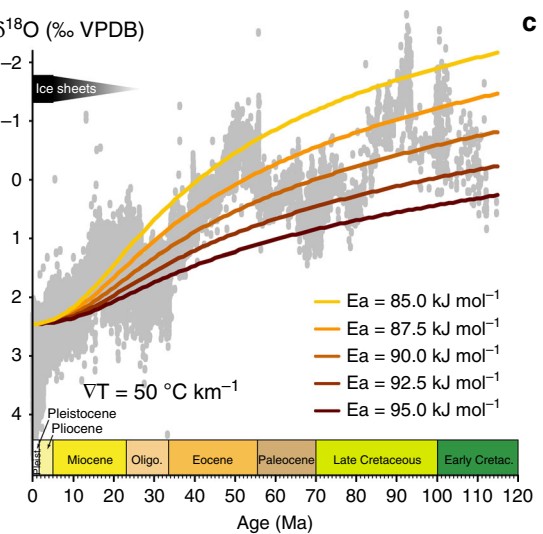

**Fig. 3** Assessment of the impact of burial-induced isotope re-equilibration on the benthic foraminifera O isotope record. **a** Five-point moving average of the burial depths at the DSDP–ODP–IODP sites from which the most comprehensive O isotope compilation is principally derived[9]. The red line indicates the depth profile used for the numerical simulations. The corresponding sediment temperatures are indicated in purple. **b**, **c** The benthic foraminifera O isotope compilation from Friedrich et al.[9] (grey) is shown with the results from numerical simulations run with a diffusion activation energy of 90 kJ mol⁻¹ and a range of geothermal gradients in **b**, and with a geothermal gradient of 50 °C km⁻¹ and a range of diffusion activation energies in **c**. These curves represent the present-day O isotope compositions of benthic foraminifera tests that underwent diffusion-controlled O isotope re-equilibration during burial, assuming an ice-free ocean ($\delta^{18}O = −1‰$ VSMOW) and a temperature of 3.5 °C at the water–sediment interface

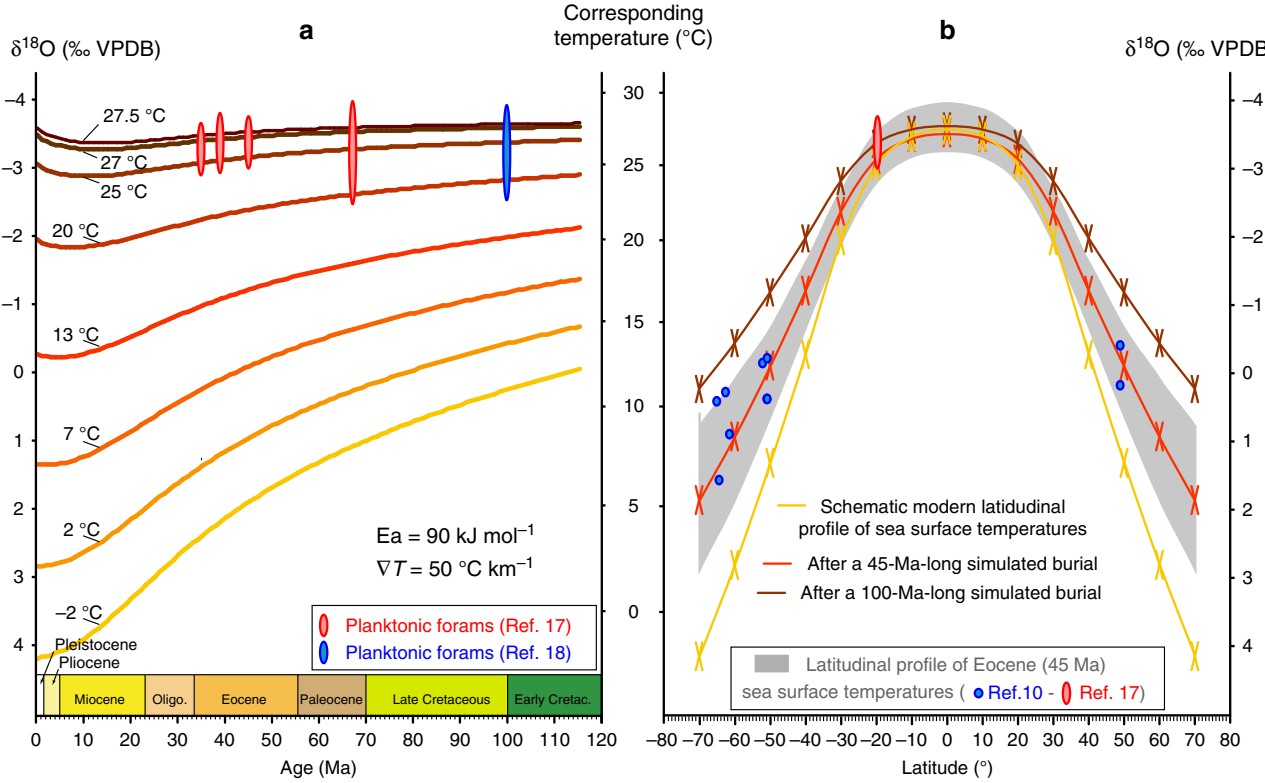

**Fig. 4** Assessment of the impact of burial-induced O isotope re-equilibration on the planktonic foraminifera O isotope records. **a** Planktonic foraminifera O isotope data from Pearson et al.[17] (Tanzania, Paleolatitude ~20° S) and from Wilson and Norris [18] (ODP Site 1052, Paleolatitude ~20° N) are shown together with the results from numerical simulations run with a geothermal gradient of 50 °C km$^{-1}$ and a diffusion activation energy of 90 kJ mol$^{-1}$, assuming an ice-free ocean ($\delta^{18}O = -1$‰ VSMOW) and a temperature of 3.5 °C at the water–sediment interface. The curves represent the present-day O isotope compositions of fossil planktonic foraminifera tests formed in equilibrium with seawaters at −2, 2, 7, 13, 20, 25, 27 and 27.5 °C. These temperatures roughly correspond to those of surface waters in the modern ocean at 70, 60, 50, 40, 30, 20, 10 and 0° latitude[10]. **b** Latitudinal profiles of the O isotope compositions of the planktonic foraminifera for different simulated burial durations: a latitude gradient similar to the modern one (yellow curve) will flatten substantially after 45 Myr (red curve) and 100 Myr (brown curve) in response to the burial-induced O isotope re-equilibration. Also shown are the Eocene (45 Ma) tropical planktonic foraminifera O isotope data from Pearson et al.[17] (red) and the high-latitude planktonic foraminifera O isotope data from Zachos et al. [10] (grey). The temperature conversion is from Anderson and Arthur [3] and assumes an ice-free ocean with a seawater $\delta^{18}O = -1$ ‰ (VSMOW)

seawater at 3.5 °C. Simulations show that burial-induced O isotope re-equilibration through diffusion can cause a significant shift in the O isotope compositions of fossil benthic foraminifera tests on a time scale on the order of $10^7$ years (>0.1‰, i.e. larger than twice the typical standard deviation of $\delta^{18}O$ measurements) (Fig. 3b, c). Worse, foraminifera tests that formed 100 Myr ago in equilibrium with a seawater at 3.5 °C would today have experienced isotope re-equilibration equivalent to a decrease in their bulk $\delta^{18}O$ of about 3‰ (Fig. 3b, c). The bulk O isotope compositions of these re-equilibrated foraminifera tests could be interpreted to indicate a seawater about 15 °C warmer. Simulations thus show that burial-induced isotope re-equilibration of foraminifera tests leads to substantial paleotemperature over-estimations. Our results suggest that instead of indicating a global cooling of the deep ocean during the late Cretaceous and Paleogene, the benthic foraminifera O isotope record principally reflects burial-induced isotope re-equilibration through solid-state diffusion. In other words, the late Cretaceous and Paleogene deep oceans were likely much colder than is currently thought.

Because the sediment stack heats up, the diffusive isotope re-equilibration impacts the O isotope composition of the fossil foraminifera tests that formed in cold waters (i.e. benthic species and high-latitude planktonic species) more than those that formed in warm waters (i.e. tropical planktonic species), such as

is shown by the slopes of the curves in Fig. 4a. The paleotemperatures interpreted from the cold-water foraminifera species are therefore more prone to be overestimated than the paleotemperatures from tropical planktonic species. Numerical simulations show that burial-induced O isotope re-equilibration significantly flattens the inferred paleo-latitudinal temperature gradient on a time scale of 45 Myr (Fig. 4b). Corrected for isotope re-equilibration, a steeper temperature gradient between low and high-latitude surface-ocean waters is re-established for the late Cretaceous and the Paleogene, i.e. a gradient similar to the modern one. This resolves the paradox of the late Cretaceous and the Paleogene low equator-to-pole surface-ocean thermal gradient and is consistent with climate and ocean circulation models[19,20].

In conclusion, accounting for the diffusion-controlled burial-induced O isotope re-equilibration of fossilised benthic foraminifera removes the requirement for a strong, continuous and global cooling of the deep-ocean (on the order of 15 °C) during the late Cretaceous and the Paleogene. Furthermore, the present study suggests that the vertical and latitudinal temperature gradients of the late Cretaceous and Paleogene oceans were likely not very different from the current ones. Importantly, because O isotope re-equilibration through solid-state diffusion is a slow process, it likely had little impact on recent (<10 Ma) high frequency signals, such as the glacial to interglacial fluctuations[33]

(driven by oscillations of Earth's orbit and mainly related to fluctuations of the seawater O isotope composition). However, these processes have potentially attenuated the relative amplitudes of older, transient signals, such as the Eocene Oligocene transition or the Palaeocene Eocene thermal maximum[7–9].

## Methods

**Isotope re-equilibration experiments**. A synthetic pure $^{18}$O seawater solution was prepared by adding NaCl and NaHCO$_3$ to pure H$_2$$^{18}$O until reaching 0.55 and 0.003 mol L$^{-1}$, respectively. Sub-modern planktonic foraminifera (*G. bulloides*) from modern sediments of the Gulf of Lion, France, with bulk $\delta^{18}$O ~ 1.35 ± 0.05‰ (VPDB) were rinsed three times in ethanol using an ultrasonic bath. Gold capsules, each containing 160 μg of cleaned foraminifera tests (~12 specimens) and 100 μL of this synthetic seawater solution, were sealed using a Lampert PUK 4 welding machine and placed in Parr autoclaves for 82 days at $T = 300\,°C$ and $P = 200$ bars. Note that with a $^{18}$O/($^{16}$O+$^{18}$O) ratio of the foraminifera test close to 0 (because of the rarity of $^{18}$O) when the ratio of the solution is close to 1, the mass balance is such that the water $\delta^{18}$O can be considered constant during the experiments, regardless of the magnitude of the isotope exchange during the experiments.

Speciation calculations were performed by running the geochemical code CHESS[34] using the thermodynamic database of the EQ3/6 code[35]. Activity coefficients for aqueous species were calculated using the Davies equation[36]; the electrical balance was achieved using the H$^+$ concentration. At ambient temperatures (293 K), the solution is slightly undersaturated with respect to calcite (within uncertainty bounds). At this temperature, assuming that the dissolution of calcite is thermodynamically controlled (i.e. neglecting kinetic barriers), equilibrium would be reached after the dissolution of <2% of the tests. At the experimental temperature (573 K), because of the retrograde solubility of calcite, saturation with respect to calcite is reached for a dissolution progress corresponding to only 0.1% of the tests. Therefore, we can assume that the NanoSIMS images reflect an isotope re-equilibration very close to chemical equilibrium, with little or no contributions from secondary crystallisation from the bulk solution.

Because the carbonate 'building blocks' of planktonic and benthic foraminifera are the same at microscales[29,30], we confidently use the results of experiments conducted on planktonic foraminifera as representing the results for both planktonic and benthic foraminifera.

**Characterisation techniques**. SEM observations were performed on foraminifera tests deposited on aluminium stubs and with 5-nm thick coatings of gold using a SEM-FEG Ultra 55 Zeiss (IMPMC, Paris, France) microscope operating at a 2-kV accelerating voltage and a working distance of 2 mm for imaging with secondary electrons and at a 15-kV accelerating voltage and a working distance of 7.5 mm for imaging with backscattered electrons and EDXS mapping.

Isotope maps were produced on triplicate samples with the Cameca NanoSIMS 50 (IMPMC, Paris, France). A thorough technical explanation of the NanoSIMS instrument has been provided in a recent review[23]. Briefly, the NanoSIMS ion microprobe is a secondary ion mass spectrometry instrument characterised by an extremely high spatial resolution, a high sensitivity, a high mass-resolving power and a multicollection capability[34]. Using a focused primary beam of $^{133}$Cs$^+$ ions, secondary ions were sputtered from the sample surface, typically to a depth of ~100 nm. $^{18}$O$^-$ and $^{16}$O$^-$ ions from the sample were simultaneously detected (multicollection mode) by electron multipliers at a mass-resolving power of ~9000 ($M/\Delta M$). At this mass-resolving power, the measured secondary ions were resolved from the potential interferences. Images were obtained from a presputtered surface area (the same surface area previously imaged using SEM) by rastering the primary beam across the sample surface. The primary beam was focused to a spot size of ~150 nm, and the pixel size was adjusted so that it was smaller than the size of the primary beam. An electron gun supplied the electrons to the sputtered surface during the analysis to compensate for positive charge deposition from the primary beam and to minimise the surface charging effects. Imaging data were processed using custom-made software (LIMAGE, L. Nittler, Carnegie Institution of Washington).

**Modelling the experiments**. In the present experiments, the driving force for the isotope re-equilibration through solid-state diffusion is the large $^{18}$O excess of the fluid. The most generic expression describing the temperature dependence of the diffusion coefficient can be written as follows:

$$D = D_0 \exp\left(-\frac{\mathrm{Ea}}{RT}\right), \tag{1}$$

where $D_0$ is the intrinsic diffusion constant, Ea is the activation energy of oxygen diffusion in calcite, $R$ is the ideal gas constant and $T$ is the absolute temperature. Although the bulk diffusion of oxygen in calcite is the sum of the contributions of both the grain boundary and volume diffusion, it is commonly assumed that the volume diffusion is slow and can be ignored[37,38].

Applied to the present case, Eq. (1) becomes:

$$D_{\mathrm{foram}} = D_{0,\mathrm{foram}} \exp\left(-\frac{\mathrm{Ea}_{\mathrm{foram}}}{RT_{\mathrm{xp}}}\right), \tag{2}$$

where $T_{\mathrm{xp}}$ is the absolute temperature of the experiment and Ea$_{\mathrm{foram}}$ is the activation energy of oxygen solid-state diffusion in foraminifera calcite. No consensus exists in the literature on the value of this last parameter (see below).

In the present experiments, the initial and boundary conditions can be expressed as follows:

$$C(x, t = 0) = 0, \; \forall x > 0 \tag{3}$$

$$C(x = 0, t) = C_0 = 1 \tag{4}$$

with $C$ standing for the $^{18}$O concentration (i.e. the molar $^{18}$O/($^{16}$O+$^{18}$O) ratio) in foraminifera calcite, $t$ is time and $x$ is the length (positive distance from the fluid/solid interface into the solid).

The diffusion equation for $C$ reads as follows (Fick's second law):

$$\frac{\partial C}{\partial t} = D_{0,\mathrm{foram}} \exp\left(-\frac{\mathrm{Ea}_{\mathrm{foram}}}{RT_{\mathrm{xp}}}\right)\frac{\partial^2 C}{\partial x^2}, \tag{5}$$

According to Crank[39], Because the calcite fraction quantitatively affected by the diffusion process is generally small compared to the total calcite size/volume, the analytical solution for the diffusion into an infinite one-dimensional medium can be applied to calculate the $^{18}$O concentration along the profile after the experimental duration $t_{\mathrm{xp}}$:

$$C(x) = C_0 \left(1 - \mathrm{erf}\left(x \big/ \left(2\sqrt{D_{0,\mathrm{foram}}\exp\left(-\frac{\mathrm{Ea}_{\mathrm{foram}}}{RT_{\mathrm{xp}}}\right)t_{\mathrm{xp}}}\right)\right)\right). \tag{6}$$

Because the error function is defined as follows:

$$\mathrm{erf}(y) = \frac{2}{\sqrt{\pi}}\int_0^y \exp(-\tau^2)\,\mathrm{d}\tau, \tag{7a}$$

with

$$y = \frac{x}{2\sqrt{D_{0,\mathrm{foram}}\exp\left(-\frac{\mathrm{Ea}_{\mathrm{foram}}}{RT_{\mathrm{xp}}}\right)t_{\mathrm{xp}}}}, \tag{7b}$$

its derivative can be written as follows:

$$\frac{\partial}{\partial y}\left(\mathrm{erf}(y)\right) = \frac{2}{\sqrt{\pi}}\exp(-y^2). \tag{8}$$

The flux of matter $F$ entering the calcite domain at the interface $x = 0$ can be expressed as follows:

$$F(0, t) = -D_{\mathrm{foram}}\frac{\partial C}{\partial x}, \tag{9}$$

with (according to Eqs. (6) and (8)):

$$\frac{\partial C}{\partial x} = \frac{\partial C}{\partial y}\frac{\partial y}{\partial x} = \frac{-C_0 \exp(-y^2)}{\sqrt{\pi D_{\mathrm{foram}} t}}, \tag{10}$$

leading to the following expression for the flux $F$ at the interface $x = 0$:

$$F(0, t) = \frac{C_0 \sqrt{D_{\mathrm{foram}}}}{\sqrt{\pi t}}, \tag{11}$$

The cumulative flux ($F_c$), which represents the amount of $^{18}$O that has entered the calcite domain during the experimental duration $t_{\mathrm{xp}}$, can be expressed as follows:

$$F_c\left(0, t_{\mathrm{xp}}\right) = \int_0^{t_{\mathrm{xp}}} F(0, \tau)\,\mathrm{d}\tau = C_0 \frac{\sqrt{D_{\mathrm{foram}}}}{\sqrt{\pi}}\int_0^{t_{\mathrm{xp}}} \frac{1}{\sqrt{\tau}}\,\mathrm{d}\tau = 2C_0 \frac{\sqrt{D_{\mathrm{foram}} t_{\mathrm{xp}}}}{\sqrt{\pi}}. \tag{12}$$

From a physical standpoint, this result illustrates that the integral of any diffusion profile that satisfies the abovementioned boundary conditions is equivalent to that of a step function of the height $C_0$ with a length $d$ of:

$$d = \frac{2}{\sqrt{\pi}}\sqrt{D_{\mathrm{foram}} t_{\mathrm{xp}}}. \tag{13}$$

Because the initial $^{18}$O/($^{16}$O+$^{18}$O) value of the foraminifera is close to 0, the $^{18}$O/($^{16}$O+$^{18}$O) value of the foraminifera calcite crystals at the end of the experiment ($Q$) is equivalent to the ratio of the re-equilibrated volume to the initial volume. Assuming that the foraminifera calcite crystals have a simple spherical geometry with a radius $r_0$ and that $d \ll r_0$ (as suggested by NanoSIMS imaging),

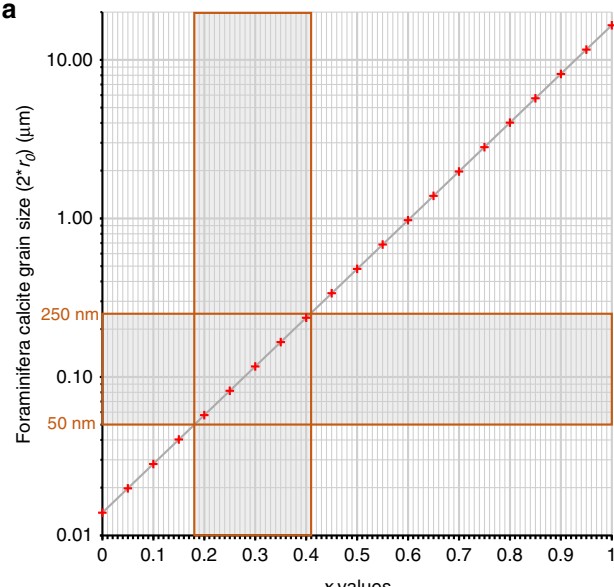

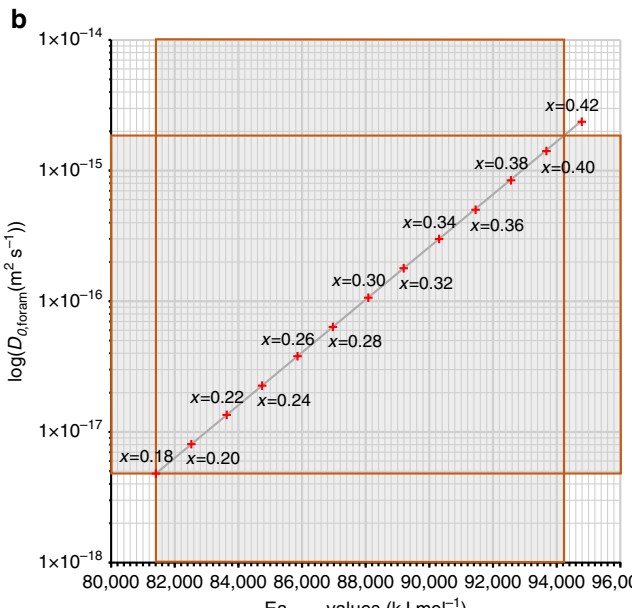

**Fig. 5** Determination of the $Ea_{foram}$ and $D_{0,foram}$ values for the oxygen solid-state diffusion in foraminifera calcites. **a** Foraminifera calcite grain sizes ($2*r_0$ values) obtained using Eq. (15) for the different $x$ values (Eq. (16)). **b** $D_{0,foram}$ and $Ea_{foram}$ values obtained using Eq. (16) for the different $x$ values

$Q$ can be written as follows:

$$Q = \frac{\frac{4}{3}\pi r_0^3 - \frac{4}{3}\pi (r_0 - d)^3}{\frac{4}{3}\pi r_0^3}. \tag{14}$$

Combining Eqs. (2), (13) and (14), the mean radius of the calcite grains ($r_0$) can thus be expressed as follows:

$$r_0 = \frac{d}{1 - (1-Q)^{1/3}} = \frac{2}{\sqrt{\pi}} \frac{\sqrt{D_{foram} t_{xp}}}{1 - (1-Q)^{1/3}} = \frac{2}{\sqrt{\pi}} \frac{\sqrt{D_{0,foram} \exp\left(-\frac{Ea_{foram}}{RT_{xp}}\right) t_{xp}}}{1 - (1-Q)^{1/3}}. \tag{15}$$

Given the actual calcite grain size of foraminifera (50–250 nm)[29,30], i.e. $r_0$ values of 25–125 nm, the quantity of $^{18}O$ that diffused within the foraminifera tests during the experiments can be used to estimate the activation energy and the intrinsic diffusion coefficient of oxygen solid-state diffusion in foraminifera calcite ($Ea_{foram}$ and $D_{0,foram}$). A large range of several tens of kJ mol$^{-1}$ exists in the literature for the activation energy of oxygen solid-state diffusion in calcite aggregates. Farver and

Yund[27] estimated Ea values of $127 \pm 17$ kJ mol$^{-1}$ ($D_{0,Farver} = 7.6*10^{-9}$ m$^2$ s$^{-1}$), while Anderson[26] suggested that diffusion within calcites can even proceed with an activation energy as low as 70 kJ mol$^{-1}$ ($D_{0,Anderson} = 4.6*10^{-20}$ m$^2$ s$^{-1}$) at low temperatures. Applied to the present experiments, Eq. (15) yields $r_0$ values of 7 nm (i.e. a calcite grain size of ~14 nm, assuming spherical grains) with the values reported by Anderson[26], while an $r_0$ value of 8.25 μm (i.e. a calcite grain size of ~16.5 μm) is obtained when using the values reported by Farver and Yund[27]. This discrepancy from the actual grain size of foraminifera is easily explained by (1) the large uncertainty (typically several tens of kJ mol$^{-1}$) in the experimental determination of the Ea values and (2) the large dispersion of foraminifera grain sizes and effective grain boundaries, which require intermediate values of $Ea_{foram}$ and $D_{0,foram}$ computed as weighted average defined by:

$$\begin{cases} Ea_{foram} = x Ea_{Farver} + (1-x) Ea_{Anderson} \\ \log(D_{0,foram}) = x \log(D_{0,Farver}) + (1-x) \log(D_{0,Anderson}) \end{cases}, \tag{16}$$

with $x$ varying between 0 and 1.

Figure 5 shows that $x$ has to be between 0.18 and 0.41 to yield foraminifera calcite grain size values between 50 and 250 nm (i.e. $r_0$ values of 25–125 nm). This range corresponds to $Ea_{foram}$ values ranging from 81 to 94 kJ mol$^{-1}$ and $D_{0,foram}$ values ranging from $4.8 \times 10^{-18}$ to $1.83 \times 10^{-15}$ m$^2$ s$^{-1}$.

**Modelling natural settings.** We developed a numerical model to estimate the present-day $\delta^{18}O$ of fossil tests of benthic foraminifera assumed to have formed during the Cretaceous and the Cenozoic in equilibrium with a seawater/sediment interface at 3.5 °C and that underwent partial isotope re-equilibration via solid-state diffusion during sediment burial. This model was also used to estimate the present-day $\delta^{18}O$ values of fossil tests of planktonic foraminifera assumed to have formed during the Cretaceous and the Cenozoic at different latitudes, i.e. at equilibrium with seawater at different temperatures, and that underwent a similar burial history.

Because the volumes impacted by re-equilibration may not be negligible compared to the size of the foraminifera calcite building blocks, it is preferable to solve the diffusion equation with a spherical geometry rather than with a simpler 1D semi-infinite medium. Because the spherical diffusion equation expressed below does not have a simple analytical solution, the diffusion equation was numerically solved using the finite difference with an implicit time discretisation.

The driving force for isotope re-equilibration through solid-state diffusion over geological times within sediments is the temperature-dependent O isotope fractionation between foraminifera test calcite and seawater at thermodynamic equilibrium ($\Delta_{cc-sw}(\theta)$), where $\theta$ stands for the temperature in °C. A number of formulations of $\Delta_{cc-sw}(\theta)$ can be found in the literature[2]. The most widely used equation is the one reported by Anderson and Arthur[3]), which relates $\Delta_{cc-sw}$ and $\theta$ as follows:

$$\theta = 16.0 - 4.14(\Delta_{cc-sw}) + 0.13(\Delta_{cc-sw})^2, \tag{17}$$

with

$$\Delta_{cc-sw}(\theta) = \delta^{18}O_{cc} - \delta^{18}O_{sw}, \tag{18}$$

where $\delta^{18}O_{cc}$ is the O isotope composition of the test calcite (in ‰, relative to VPDB) and $\delta^{18}O_{sw}$ is the O isotope composition of seawater (in ‰, relative to VSMOW). This equation allows the $\delta^{18}O_{cc}$ to be written as a function of $\delta^{18}O_{sw}$ and $T$ (in K):

$$\delta^{18}O_{cc} = \frac{4.14 - \sqrt{(4.14)^2 - 4*0.13*(16 - T + 273.15)}}{2*0.13} + \delta^{18}O_{sw}. \tag{19}$$

Assuming spherical calcite crystals, the mathematical model describing diffusion in spherical coordinates is:

$$\frac{\partial C}{\partial t} = D_{0,foram} \exp\left(-\frac{Ea_{foram}}{RT}\right) \frac{1}{r^2} \frac{\partial}{\partial r}\left(r^2 \frac{\partial}{\partial r}\right), \tag{20}$$

with the following initial and boundary conditions in natural settings:

$$\delta^{18}O_{cc}(r, t=0) = \Delta_{cc-sw}(T(t=0)) + \delta^{18}O_{sw}(t=0), \forall r \in [0; r_0], \tag{21}$$

$$\delta^{18}O_{cc}(r=r_0, t) = \Delta_{cc-sw}(T(t)) + \delta^{18}O_{sw}(t), \tag{22}$$

where $r$ represents the distance to the solid centre of the spherical foraminifera calcite crystals and $r_0$ represents their radii. Equation (20) was numerically solved using the implicit finite differences method[40] with M cells numbered from the solid/fluid interface to the solid centre. The numerical discretisation leads to the following mass balance equation:

$$-\frac{S_{i,i-1}D(z)}{dr}\delta^{18}O_{cc_{i-1}}^{n+1} + \left(\frac{V_i}{\Delta t} + \frac{S_{i,i-1}D(z)}{dr} + \frac{S_{i,i+1}D(z)}{dr}\right)\delta^{18}O_{cc_i}^{n+1}$$
$$-\frac{S_{i,i+1}D(z)}{dr}\delta^{18}O_{cc_{i+1}}^{n+1} = \frac{V_i}{\Delta t}\delta^{18}O_{cc_i}^n, \tag{23}$$

where $S_{i,i-1} = 4\pi(r - dr/2)^2$ (respectively $S_{i,i+1}$) is the surface area between the $i$th and $(i-1)$th (respectively the $i$th and $(i+1)$th) spherical shells of calcite with thicknesses of $dr$, located at a distance $r-dr$ from the centre. $\delta^{18}O_{cc_{i-1}}^{n+1}$, $\delta^{18}O_{cc_i}^{n+1}$ and $\delta^{18}O_{cc_{i+1}}^{n+1}$ represent the $^{18}O$ concentrations within the $(i-1)$th, $i$th and $(i+1)$th shells at time step $n+1$; and $\delta^{18}O_{cc_i}^n$ is the $^{18}O$ concentration within the $i$th shell at the time step $n$. $V_i$ is the volume of the $i$th shell defined as $\frac{4}{3}\pi((r + dr/2)^3 - (r - dr/2)^3)$, $\Delta t$ is the time interval and $D(z)$ is the diffusion coefficient at a given burial depth $z$.

Applying the appropriate boundary conditions permits the calculation of the mass balance for the cell of the mesh in contact with the pore water:

$$V_1 \frac{\delta^{18}O_{cc_1}^{n+1} - \delta^{18}O_{cc_1}^n}{\Delta t} = S_0 D(z) \frac{\delta^{18}O_{cc_{in}}^{n+1} - \delta^{18}O_{cc_1}^{n+1}}{dr} + S_{1,2} D(z) \frac{\delta^{18}O_{cc_2}^{n+1} - \delta^{18}O_{cc_1}^{n+1}}{dr}, \quad (24)$$

with $S_0$ representing the surface area of the calcite crystals of radius $r_0$.

Similarly, the mass balance can be calculated for the $M$th cell corresponding to the centre of a calcite sphere by supposing that the inner interface is impermeable, which sets a no flux boundary condition:

$$V_M \frac{\delta^{18}O_{cc_M}^{n+1} - \delta^{18}O_{cc_M}^n}{\Delta t} = S_{M,M-1} D(z) \frac{\delta^{18}O_{cc_{M-1}}^{n+1} - \delta^{18}O_{cc_M}^{n+1}}{dr}. \quad (25)$$

Finally, the overall present-day isotope composition of the foraminifera ($\delta^{18}O_{cc}^{overall}(t_{age})$) is calculated via the cumulative isotope composition of calcite on each cell in the domain ($t_{age}$ being the time needed to reach present day):

$$\delta^{18}O_{cc}^{overall}(t_{age}) = \frac{\sum_{i=1}^{M} V_i \delta^{18}O_{cc_i}(t_{age})}{(4/3)\pi r_0^3}. \quad (26)$$

**Input parameters of the simulations.** The main parameters of the present simulations were the seawater temperature, the O isotope composition of seawater, the sediment burial rate, the geothermal gradient and the activation energy of oxygen diffusion in foraminifera calcite (as demonstrated below, the present simulations do not depend on the intrinsic diffusion coefficient).

Here, benthic foraminifera were assumed to have grown in equilibrium with the contemporary deep seawater at a constant temperature of 3.5 °C, and the planktonic foraminifera were assumed to have grown in equilibrium with the surface seawaters at $-2$, 2, 7, 13, 20, 25, 27 and 27.5 °C, roughly corresponding to the surface waters of the modern ocean at 70, 60, 50, 40, 30, 20, 10 and 0° latitude[10]. In other words, we assumed that the ocean temperatures remained constant during the late Cretaceous and the entire Cenozoic.

The O isotope composition of seawater has varied through geologic times, primarily as a consequence of ice sheet growth and decay[6–9]. The calculation of the influence of the global ice volume on the average seawater $\delta^{18}O$ depends on how much continental ice was on the planet at a given time. Zachos et al.[7] suggested that the growth of the ice sheets on Antarctica and in the Northern Hemisphere have been responsible for a total decline of ~2.3‰ of the seawater $\delta^{18}O$ values (1.2‰ for Antarctic ice sheets and 1.1‰ for Northern Hemisphere ice sheets). However, a value of $-1.2$‰ for an ice-free ocean is commonly encountered in the literature[2]. Models of present-day ice sheet growth have indicated that an ice-free ocean would have an average $\delta^{18}O$ value between $-0.89$ and $-1.1$‰ (VSMOW)[41,42]. Assuming a more or less ice-free ocean over the late Cretaceous and the entire Cenozoic, the $\delta^{18}O$ of the seawater was fixed to $-1$‰ (VSMOW) for the present simulations. The sediment pore water in which the foraminifera have been fossilised was assumed to be isotopically similar to the seawater in which they lived[32] (as supported by the $\delta^{18}O$ close to $-1$‰ (VSMOW) of the pore water in the old sediments).

To properly estimate the burial rate and consider the inevitable sediment compaction, the burial depth of the sediments from the DSDP/ODP/IODP sites used in the Friedrich et al.[9] compilation of the deep sea benthic $\delta^{18}O$ values have been plotted against their ages, excluding sites that have undergone significant erosion. A sliding five-point average of these data is reported in Fig. 3a with a reasonable fit with $z_{max} = 500$ m, $\alpha = 10$ and $\beta = 1.5$:

$$z = z_{max}\left(1 - \exp\left(-\frac{t}{\alpha}\right)\right)^{\beta}. \quad (27)$$

The geothermal gradients in oceanic sediments have been estimated to range between 30 and 70 °C km$^{-1}$ [32]. Because the temperature experienced by the fossil foraminifera directly controls the extent of solid-state diffusion, the present model was computed using different geothermal gradients, ranging from 40 to 60 °C km$^{-1}$. The temperature experienced by the sediments $\theta$ (in °C) is the product of the burial depth $z$ (in metres) and the geothermal gradient $\nabla\theta$ (in °C m$^{-1}$):

$$\theta = \nabla\theta * z \left(1 - \exp\left(-\frac{t}{\alpha}\right)\right)^{\beta}. \quad (28)$$

As detailed above, Ea$_{foram}$ values ranging from 81 to 94 kJ mol$^{-1}$ had to be considered. The present numerical simulations were thus conducted using different activation energies ranging from 85 to 95 kJ mol$^{-1}$.

**Non-dependence on the intrinsic diffusion coefficient.** The present numerical simulations do not depend on the intrinsic diffusion coefficient ($D_{0,foram}$), as demonstrated here. If calcite crystals are approximated by simple rods, the $Q$ ratio corresponds to:

$$Q = \frac{d}{r_0} = \frac{2}{r_0\sqrt{\pi}}\sqrt{D_{foram}t_{xp}}, \quad (29)$$

where $r_0$ is the radius of the calcite domain. Combining Eq. (29) with Eqs. (2) and (13) yields the following:

$$r_0 = \frac{2}{Q\sqrt{\pi}}\sqrt{D_{0,foram}\exp\left(-\frac{Ea_{foram}}{RT_{xp}}\right)t_{xp}}. \quad (30)$$

As emphasised here, the integral of any diffusion profile can be approximated via a step function with length $d$ and height $C_0$. Therefore, at any time, the instantaneous isotope composition of the calcites ($\delta^{18}O_{cc}(t)$) can be expressed using the following mass balance:

$$\delta^{18}O_{cc}(t) = \frac{\delta^{18}O_{cc}(r = r_0, t)d + \delta^{18}O_{cc}(r, t = 0)(r_0 - d)}{r_0}, \quad (31)$$

which yields, after rearrangement,

$$\delta^{18}O_{cc}(t) = \frac{d}{r_0}\left(\delta^{18}O_{cc}(r_0, t) - \delta^{18}O_{cc}(r, 0)\right) + \delta^{18}O_{cc}(r, 0). \quad (32)$$

Therefore, after a given duration $t_{age}$, the isotope composition of the foram ($\delta^{18}O_{cc}^{overall}(t_{age})$) as previously defined is as follows:

$$\delta^{18}O_{cc}^{overall}(t_{age}) = \frac{1}{t_{age}}\frac{1}{r_0}\int_0^{t_{age}} d(\tau)\left(\delta^{18}O_{cc}(r_0, \tau) - \delta^{18}O_{cc}(r, 0)\right)d\tau + \delta^{18}O_{cc}(r, 0). \quad (33)$$

Replacing $r_0$ and $d(\tau)$ with the values taken from Eqs (2), (13) and Eq. (30) yields the following:

$$\delta^{18}O_{cc}^{overall}(t_{age}) = \frac{1}{t_{age}}\frac{1}{\frac{2}{Q\sqrt{\pi}}\sqrt{D_{0,foram}\exp\left(-\frac{Ea_{foram}}{RT_{xp}}\right)t_{xp}}} \int_0^{t_{age}} \frac{2}{\sqrt{\pi}}\sqrt{D_{0,foram}\exp\left(-\frac{Ea_{foram}}{RT_{xp}}\right)}\left(\delta^{18}O_{cc}(r_0, \tau) - \delta^{18}O_{cc}(r, 0)\right)d\tau + \delta^{18}O_{cc}(r, 0), \quad (34)$$

which yields, after rearrangement,

$$\delta^{18}O_{cc}^{overall}(t_{age}) = \frac{1}{t_{age}}\frac{1}{\frac{1}{Q}\sqrt{\exp\left(-\frac{Ea_{foram}}{RT_{xp}}\right)t_{xp}}} \int_0^{t_{age}}\sqrt{\exp\left(-\frac{Ea_{foram}}{RT(\tau)}\right)}\left(\delta^{18}O_{cc}(r_0, \tau) - \delta^{18}O_{cc}(r, 0)\right)d\tau + \delta^{18}O_{cc}(r, 0). \quad (35)$$

Thus, $\delta^{18}O_{cc}^{overall}(t_{age})$ does not depend on $D_{0,foram}$.

This result holds true in the case of spherical calcites, at least when the characteristic length of diffusion is negligible compared to the calcite grain size, which is the case for most of the present simulations. In fact, rewriting Eq. (13) and Eq. (15) yields the following:

$$d = \frac{2}{\sqrt{\pi}}\sqrt{D_{0,foram}\exp\left(-\frac{Ea_{foram}}{RT(t)}\right)t} = \sqrt{D_{0,foram}}f(t) \quad (36)$$

and

$$r_0 = \frac{2}{\sqrt{\pi}}\frac{\sqrt{D_{0,foram}\exp\left(-\frac{Ea_{foram}}{RT_{xp}}\right)t_{xp}}}{1 - (1 - Q)^{1/3}} = \sqrt{D_{0,foram}}Cte_{xp}, \quad (37)$$

in which all the constant parameters of Eq. (15) but $\sqrt{D_{0,foram}}$ are included in $Cte_{xp}$.

If the characteristic length of diffusion is negligible compared to the calcite grain size, the instantaneous isotope composition of calcite ($\delta^{18}O_{cc}(t)$) can be written as follows:

$$\delta^{18}O_{cc}(t) = \frac{\delta^{18}O_{cc}(r_0, t)V_{shell} + \delta^{18}O_{cc}(r, 0)V_{int}}{V_0}$$
$$= \frac{\delta^{18}O_{cc}(r_0, t)4\pi(r_0 - d)^2 d + \delta^{18}O_{cc}(r, 0)\frac{4\pi}{3}(r_0 - d)^3}{\frac{4\pi}{3}r_0^3}, \quad (38)$$

where $V_{shell}$ stands for a fully re-equilibrated external shell of calcite crystals, $V_{int}$ is the internal volume unaffected by the re-equilibration and $V_0$ is the initial volume.

Replacing $d$ and $r_0$ by their values yields the following:

$$\delta^{18}O_{cc}(t) = \frac{\delta^{18}O_{cc}(r_0, t) 4\pi \left(\sqrt{D_{0,foram}}Cte_{xp} - \sqrt{D_{0,foram}}f(t)\right)^2 \sqrt{D_{0,foram}}f(t) + \delta^{18}O_{cc}(r, 0)\frac{4\pi}{3}\left(\sqrt{D_{0,foram}}Cte_{xp} - \sqrt{D_{0,foram}}f(t)\right)^3}{\frac{4\pi}{3}\left(\sqrt{D_{0,foram}}Cte_{xp}\right)^3},$$

(39)

which yields, after rearrangement,

$$\delta^{18}O_{cc}(t) = \frac{\left(\delta^{18}O_{cc}(r_0, t)\left(Cte_{xp} - f(t)\right)^2 f(t) + \frac{1}{3}\delta^{18}O_{cc}(r, 0)\left(Cte_{xp} - f(t)\right)^3\right)}{\frac{1}{3}Cte_{xp}^3}.$$

(40)

After a given duration $t_{age}$, the isotope composition of a foraminifera ($\delta^{18}O_{cc}^{overall}\left(t_{age}\right)$) is written as follows:

$$\delta^{18}O_{cc}^{overall}\left(t_{age}\right) = \frac{3}{t_{age}Cte_{xp}^3}\int_0^{t_{age}}\left(\delta^{18}O_{cc}(r_0, \tau)\left(Cte_{xp} - f(t)\right)^2 f(t) + \frac{1}{3}\delta^{18}O_{cc}(r, 0)\left(Cte_{xp} - f(t)\right)^3\right)d\tau.$$

(41)

Thus, $\delta^{18}O_{cc}^{overall}\left(t_{age}\right)$ does not depend on $D_{0,foram}$ in the case of spherical calcites.

**Data availability**. Original data and numerical codes are available from the corresponding author upon request.

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

## Acknowledgements

This work was supported by the European Research Council Advanced Grant 246749 BIOCARB to A.M. The SEM facility of the IMPMC is supported by Region Ile de France grant SESAME NI-07-593/R, INSU-CNRS, INP-CNRS, UPMC-Paris 6, and by the Agence Nationale de la Recherche (ANR) Grant No. ANR-07-BLAN-0124-01. The National NanoSIMS Facility at the MNHN is supported by the CNRS, the Région Île de France, the Ministère délégué à l'enseignement supérieur et à la recherche, and the MNHN. Special thanks go to O. Lengliné for stimulative discussions and to five anonymous reviewers for their constructive comments.

## Author contributions

S.B., D.D. and A.M. designed the experimental study; S.B. and D.D. performed the isotope re-equilibration experiments; S.B. and S.P. performed the SEM observations; S.B. and A.M. performed the NanoSIMS imaging; D.D. and P.A. developed analytical and

numerical models with critical inputs from S.B.; S.B., D.D. and A.M. wrote the manuscript.

## Additional information

**Competing interests:** The authors declare no competing financial interests.

