## [Peer Review File · Nature Communications]

Reviewer #1 (Remarks to the Author):

I was asked to review this manuscript for Nature several months ago. The review that I sent back to the authors contained a number of issues that I expanded upon in detail. I see that the authors decided to quickly turn the ms around for Nature Communications without considering my comments on the original submission. My review below, therefore, contains the same issues that I raised previously:

The authors have conducted a fascinating suite of experiments that explore the diffusive exchange of $^{18}\text{O}/^{16}\text{O}$ into fossil foraminifera by subjecting samples to elevated pressures and temperature to simulate geologic timescale exposure of CaCO_3 to elevated geothermal gradients in marine sedimentary columns. They then model the results to explore the impact of the geothermal gradient on oxygen isotope diagenesis to ask the question whether or not early Cenozoic paleotemperatures derived from fossil benthic foraminifera have been shifted towards warmer temperatures, thereby yielding ambient temperatures that severely overestimate actual bottom water conditions during the Eocene and Paleocene.

First, I congratulate the authors on a very interesting manuscript that utilizes NanoSIMS results to try to simulate a process that had previously only been discussed in theoretical terms. I believe that their observations of extensive diffusive exchange throughout fossil foraminifera shells is the first such data that I have seen in the published literature and as such is quite novel and important for considering the issues they raise in their manuscript.

My primary concern with the manuscript focuses on the modeling and conclusions that the authors draw on early Cenozoic temperatures based on their experimental results. These issues fall into a few categories. First, the authors make an implicit assumption that is never fully developed - that their experiment results obtained @300C and 200 bar pressure for 3 months can be used to simulate low temperature exchange that fossil foraminifera experience when exposed to temperatures that are <40C @ 300 to 500 bar for 10⁶ to 5x10⁷ years. Second, the model calculations do not simulate the correct conditions that exist at sites researchers obtain material from to reconstruct early Paleogene climate. For instance, the authors assume in their models that the geothermal gradient for the upper 1000m of the sediment column ranges from 40-60C and that samples are derived from sediment depths of up to 500m. While this geothermal gradient is correct over new oceanic crust near spreading centers, it overestimates the gradient above old/cold crust in regions that are typically cored for Paleogene paleoceanographic reconstructions (e.g. Shatsky Rise, Walvis Ridge etc.) which is typically 20-30C/km. Furthermore, one cannot simply assume that sedimentation rates have been constant for the past 50 Ma and that a sedimentation rate of 1cm/kyr can be extrapolated out for 5 x 10⁷ years. In reality, most regions have large hiatuses in the sediment column with non-existent sediment accumulation for tens of millions of years because the CCD was shallower at times in the past. Paleoceanographers search for these sites because the Eocene and Paleocene is then much closer to the sediment column surface. For this reason, Eocene and Paleocene sediment samples often come from sediment depths of only 150-300 m below the seafloor. Taken together, this means that the worst case scenario for the proposed diffusive diagenesis would be to expose fossil shells to temperatures that are at most 10C above ambient, not 40-60C.

My third issue comes from the literature. A recent study of Paleocene and Eocene foraminifera using SIMS analyses of micron scale domains within fossil planktic shells [Kozdon et al., 2011] show that components of Eocene fossil foraminifera shells are altered whereas other domains have considerably lower $\delta^{18}\text{O}$ values that are consistent with other available proxies. If the diffusive/diagenetic mechanism of the authors is correct then how do these fossil foraminifera display a range of values across a small micron scale range in the shells? There is no reason why benthic foraminifera (the target in the manuscript) should behave differently than planktic foraminifera especially given the experimental results that were conducted on Quaternary planktonic species. Wouldn't diffusion homogenize such variability rather than allowing intrashell

variability to remain in the fossils?

Finally, there is considerable evidence for an ice free Earth with warm conditions at high latitudes in the Eocene and Paleocene. Not only does fossil evidence point to a warm Earth in the terrestrial record, but organic geochemical evidence from proxies such as TEX86 agree well with existing oxygen isotope data from planktic foraminifera, and the benthic records are consistent with a thermal gradient in the ocean for an ice free world. I also add that a diagenetic mechanism such as that proposed by the authors to overprint deep ocean temperatures, should not allow us to observe rapid climatic features such as the terminal Eocene/Oligocene boundary deep ocean cooling event at 33 Ma nor obliquity/eccentricity oscillations, in the oxygen isotope record.

Although the experiments that the authors conducted are quite novel and incredibly interesting, I find that the application and extrapolation of their results to fossil Paleogene foraminifera does not utilize model conditions that likely existed in the sediment column that these fossils were exposed to. Furthermore, it is not sufficient to extrapolate the experimental results to the fossil record without trying to link experimental pressure, temperature and temporal conditions to these parameters in the natural environment. Addressing the issue of how foraminifera oxygen isotope records could be diagenetically altered when the records display clear variations on timescales <104 years (e.g. PETM $\delta^{18}\text{O}$ shift) is a critical omission. Although the experiments in this study are excellent and intriguing, I am not convinced the authors can draw the conclusions they do on Cenozoic oxygen isotope paleoceanography. For this reason, the manuscript does not rise to the level of a Nature publication and I must recommend that the manuscript be rejected. However, I encourage the authors to submit their paper to a specialty journal and focus on the mechanism rather than trying to make a sensational splash with an application that does not address the fundamental issues raised above.

Reviewer #2 (Remarks to the Author):

I have attached the review as a pdf file (using the first of the Browse buttons below because when I enter the review here, all of the formatting is lost - which makes it quite difficult to read.

It is called: Review 18O and NanoSIMS.pdf

Revision of “Burial-induced oxygen isotope re-equilibration of fossil foraminifera explains ocean paleotemperature paradoxes” by Bernard, Daval, Ackerer, Pont, and Meibom.

Response to Referees

Reviewer #1

1. The authors have conducted a fascinating suite of experiments that explore the diffusive exchange of $^{18}\text{O}/^{16}\text{O}$ into fossil foraminifera by subjecting samples to elevated pressures and temperature to simulate geologic timescale exposure of CaCO_3 to elevated geothermal gradients in marine sedimentary columns. They then model the results to explore the impact of the geothermal gradient on oxygen isotope diagenesis to ask the question whether or not early Cenozoic paleotemperatures derived from fossil benthic foraminifera have been shifted towards warmer temperatures, thereby yielding ambient temperatures that severely overestimate actual bottom water conditions during the Eocene and Paleocene.

We thank the Reviewer #1 for this positive summary of our work.

Precision: Our work addresses the Cretaceous and Cenozoic deep and surface ocean paleotemperatures derived from fossil benthic and planktonic foraminifera.

2. I was asked to review this manuscript for Nature several months ago. The review that I sent back to the authors contained a number of issues that I expanded upon in detail. I see that the authors decided to quickly turn the ms around for Nature Communications without considering my comments on the original submission. My review below, therefore, contains the same issues that I raised previously.

Actually, in addition to numerous improvements based on 4 reviews, the version submitted to Nature Communications also contains a major new scientific element and a new figure (Fig. 4) demonstrating how we explain the misconstrued idea that the Paleogene was characterized by an extremely flat thermal gradient between equator and the poles. This removes a long-standing conundrum.

3. First, I congratulate the authors on a very interesting manuscript that utilizes NanoSIMS results to try to simulate a process that had previously only been discussed in theoretical terms. I believe that their observations of extensive diffusive exchange throughout fossil foraminifera shells is the first such data that I have seen in the published literature and as such is quite novel and important for considering the issues they raise in their manuscript.

We thank the Reviewer #1 for this very positive comment.

4. My primary concern with the manuscript focuses on the modeling and conclusions that the authors draw on early Cenozoic temperatures based on their experimental results. These issues fall into a few categories. First, the authors make an implicit assumption that is never fully developed - that their experiment results obtained @300C and 200 bar pressure for 3 months can be used to simulate low temperature exchange that fossil foraminifera experience when exposed to temperatures that are <40°C @ 300 to 500 bar for 10^6 to 5×10^7 years.

We agree that scaling-up can, in general, be problematic. But, in contrast to what the referee indicates, we did not extrapolate the results of our laboratory experiments to natural settings. We used laboratory experiments to demonstrate/illustrate/visualize that O isotope signatures of fossil foraminifera are certainly impacted by diffusion and, in parallel, we calculated/quantified what occurs in natural settings based on literature information about diffusion and relevant information about sediment conditions from DSDP/ODP/IODP reports. We made this point clearer in the revised version of our manuscript (cf lines 137-141 and 453-471).

5. Second, the model calculations do not simulate the correct conditions that exist at sites researchers obtain material from to reconstruct early Paleogene climate. For instance, the authors assume in their models that the geothermal gradient for the upper 1000m of the sediment column ranges from 40-60C and that samples are derived from sediment depths of up to 500m. While this geothermal gradient is correct over new oceanic crust near spreading centers, it overestimates the gradient above old/cold crust in regions that are typically cored for Paleogene paleoceanographic reconstructions (e.g. Shatsky Rise, Walvis Ridge etc.) which is typically 20-30C/km. Furthermore, one cannot simply assume that sedimentation rates have been constant for the past 50 Ma and that a sedimentation rate of 1cm/kyr can be extrapolated out for 5×10^7 years. In reality, most regions have large hiatuses in the sediment column with non-existent sediment accumulation for tens of millions of years because the CCD was shallower at times in the past. Paleoceanographers search for these sites because the Eocene and Paleocene is then much closer to the sediment column surface. For this reason, Eocene and Paleocene sediment samples often come from sediment depths of only 150-300 m below the seafloor. Taken together, this means that the worst case scenario for the proposed diffusive diagenesis would be to expose fossil shells to temperatures that are at most 10C above ambient, not 40-60C.

We did not estimate sediment depths using unrealistic geothermal gradients or sedimentation rates. We used real depths and temperatures (as indicated on Figure 3a) from the DSDP/ODP/IODP reports for the sites from which the largest benthic foraminifera record has been constructed (Zachos et al., 2001 & 2008; Friedrich et al., 2011). Calculations were made using geothermal gradients of about $50 \text{ }^\circ\text{C.km}^{-1}$ for a burial depth never exceeding 500 m, i.e. at temperatures of about 25°C, which is certainly realistic (cf Figure below from Malinverno et al., Scientific Reports, 2015). Foraminifera never undergo temperatures of 40-60 °C, neither in natural settings nor in the present numerical simulations. We made this point clearer in the revised version of our manuscript (cf lines 137-141 and 453-471).

Figure: In situ sediment temperature measurements taken in 207 drill sites (a) and histograms of the number of temperature measurements per site (b), of the temperature at the seafloor (c), and of the geothermal gradient at each site (d). The dashed line in (a) is the linear geothermal gradient for the median gradient (0.053 °C/m) in all the sites.

Source: Malinverno et al., 2015

6. My third issue comes from the literature. A recent study of Paleocene and Eocene foraminifera using SIMS analyses of micron scale domains within fossil planktic shells [Kozdon et al., 2011] show that components of Eocene fossil foraminifera shells are altered whereas other domains have considerably lower $\delta^{18}\text{O}$ values that are consistent with other available proxies. If the diffusive/diagenetic mechanism of the authors is correct then how do these fossil foraminifera display a range of values across a small micron scale range in the shells? There is no reason why benthic foraminifera (the target in the manuscript) should behave differently than planktic foraminifera especially given the experimental results that were conducted on Quaternary planktonic species. Wouldn't diffusion homogenize such variability rather than allowing intrashell variability to remain in the fossils?

Kozdon et al. (Paleoceanography, 2011) showed that secondary calcites do not exhibit the same $\delta^{18}\text{O}$ values as the unrecrystallized domains of foraminifera tests. This is not inconsistent with our conclusions. We did not investigate here the impact of recrystallization. We certainly hope that by now all foram paleoclimatologists try meticulously to avoid recrystallized specimens. As shown by Kozdon et al. (Paleoceanography, 2011; Paleoceanography, 2013), but also by Killingley (Nature, 1983), Schrag et al. (Chemical Geology, 1999), Edgar et al. (Paleoceanography, 2013; GCA, 2015) among others, foraminifera recrystallization seriously bias paleotemperature reconstructions (cf lines 44-54).

Here, we investigated the impact of diffusion, a process that leaves no visible trace, in contrast to recrystallization. Our laboratory experiments clearly demonstrate that diffusion does not lead to isotopically homogeneous foraminifera tests, at least as long as foraminifera are not entirely re-equilibrated (which cannot happen in natural settings on relevant timescales – cf Figure 2 and lines 126-128).

In fact, because of the small size of both benthic and planktonic foraminifera calcite domains (Cuif et al., 2011) and because of the very small the characteristic diffusion length of oxygen in calcite (Anderson, 1983; Farver and Yund, 1998), diffusion leads to isotopically heterogeneous foraminifera tests at the nanoscale. This heterogeneity is also related to the distribution of grain sizes among the carbonate ‘building blocks’ of foraminifera tests (either benthic or planktonic): for a given diffusion length, smaller calcite domains will be more impacted than bigger calcite domains. Such nanoscale isotopic heterogeneities cannot be observed using a conventional ion microprobe because of the large beam spot (typically > 10 μm).

Finally, because the carbonate ‘building blocks’ of benthic and planktonic foraminifera are the same at the microscale (Cuif et al., 2011), we confidently question the Cretaceous and Cenozoic paleotemperatures derived from fossil benthic and planktonic foraminifera based on experiments performed on planktonic foraminifera (cf lines 210-213).

7. Finally, there is considerable evidence for an ice free Earth with warm conditions at high latitudes in the Eocene and Paleocene. Not only does fossil evidence point to a warm Earth in the terrestrial record, but organic geochemical evidence from proxies such as TEX86 agree well with existing oxygen isotope data from planktic foraminifera, and the benthic records are consistent with a thermal gradient in the ocean for an ice free world.

We agree that oceans were ice-free during most of the Paleogene and the Cretaceous (this is why we used a mean $\delta^{18}\text{O}$ of -1‰ for the pore water in our simulations). Yet, to be ice free, the oceans do not have to be more than a few degrees warmer than today, certainly not 20 °C warmer. In fact, an average global warming of just 2 °C compared with preindustrial temperatures would be enough to melt more than 75% of the Greenland ice sheet (Abe-Ouchi et al., Nature, 2013; Dutton et al., Science, 2015). Greenland has undergone one or more episodes of full deglaciation during the past million years even though temperatures were only a couple of degrees warmer than today (Schaefer et al., Nature, 2016). For instance, the last interglacial period was characterized by a global mean surface temperature that was 0.7 to 2 °C higher than the preindustrial state (Clark and Huybers, Nature, 2009; McKay et al., GRL, 2011) and an eustatic sea level that can only be explained by the melting of the Greenland and Antarctic ice sheets (Dutton et al., Science, 2012).

8. I also add that a diagenetic mechanism such as that proposed by the authors to overprint deep ocean temperatures, should not allow us to observe rapid climatic features such as the terminal Eocene/Oligocene boundary deep ocean cooling event at 33 Ma nor obliquity/eccentricity oscillations, in the oxygen isotope record.

As stated in our manuscript, because isotope re-equilibration through solid-state grain boundary and volume diffusion remains a slow process, it likely had little impact on recent (< 10 Ma) high-frequency signals such as the glacial to interglacial fluctuations driven by oscillations in Earth’s orbit and mainly related to fluctuations of the O isotope composition of seawater. However, isotope re-equilibration through diffusion has the capability to attenuate/smooth the relative amplitude of older, transient signals, such as the Eocene-Oligocene transition or the Paleocene-Eocene thermal maximum. Attenuating a signal is not erasing it. We made this point even clearer in the revised version of our manuscript (cf lines 175-181).

For the record, the figure below compares the compilations by Zachos et al. from 2001 in Science and from 2008 in Nature illustrates the great deal of caution that should be exercised when interpreting these so-called transient high-frequency signals: some of these signals simply appear/disappear between the two publications as a result of unexplained 'revisions' of the data and new corrections for vital effects. The differences between these two compilations are sometimes dramatic.

Comparison of compilations by Zachos et al. and published 2001 in Science and 2008 in Nature.

9. Although the experiments that the authors conducted are quite novel and incredibly interesting, I find that the application and extrapolation of their results to fossil Paleogene foraminifera does not utilize model conditions that likely existed in the sediment column that these fossils were exposed to. Furthermore, it is not sufficient to extrapolate the experimental results to the fossil record without trying to link experimental pressure, temperature and temporal conditions to these parameters in the natural environment.

See answer to point 5 above.

10. Addressing the issue of how foraminifera oxygen isotope records could be diagenetically altered when the records display clear variations on timescales $<10^4$ years (e.g. PETM $\delta^{18}O$ shift) is a critical omission.

We did not question the existence of rapid climatic events (see point 8). In fact, any of the high frequency signals that deviate from the monotonic curves generated by our numerical simulations either reflects global scale climatic events that really occurred, fluctuations of the O isotope composition of seawater related to the ice volume, or local differences between sites (cf lines 175-181).

11. Although the experiments in this study are excellent and intriguing, I am not convinced the authors can draw the conclusions they do on Cenozoic oxygen isotope paleoceanography. For this reason, the manuscript does not rise to the level of a Nature publication and I must recommend that the manuscript be rejected. However, I encourage the authors to submit their paper to a specialty journal and focus on the mechanism rather than trying to make a sensational splash with an application that does not address the fundamental issues raised above.

We are not trying to make a sensational splash. We are trying to correct the existing sensational splash that deep ocean temperatures were something like 15 to 20 °C warmer during the Cretaceous and Paleogene than they are today.

Reviewer #2

12. This paper presents some beautiful images from a cross section from one foraminifera, after exposure to a solution containing a high proportion of ^{18}O , compared with a similar section from an untreated foram. The treatment lasted 3 months at 300 °C and 200 bars. The authors used SEM images to show that there were no visible changes in the morphology of the forams after treatment and they developed a model, which is explained in the supplementary information, to show how the data can be interpreted to challenge current wisdom about palaeotemperatures.

We thank the Reviewer #2 for this positive summary.

13. The NanoSIMS images demonstrate a dramatic change in isotope ratio that shows, without doubt, that isotope ratios in solids are not constant with time. They can be reset by interaction with pore solutions where the isotope ratio is different. This is not the first evidence that I have become aware of, to make this claim. Some is published (but rarely cited; the isotope temperature proxy community is very strong) and some is not published. The main point is that isotope ratios are dynamic and cannot be considered to precisely represent conditions at the time the solids were formed. This concept challenges the status quo and that is a very good thing. There is no doubt that recrystallisation, i.e. the result of dynamic equilibrium, changes isotope ratios and can do so without leaving a visual trace in the solid. This paper provides evidence.

We thank the Reviewer #2 for this positive summary. Of note, we did not investigate the impact of recrystallization, but the impact of diffusion, a process that leaves no visible trace, in contrast to recrystallization. We made this point clearer in the revised version of our manuscript (cf lines 97-110).

14. It is very important that this work be published - but there are some important changes that must be made first. There are many examples of sloppiness, which is not consistent with submission to a high level journal aimed at a general scientific audience. The arguments in the text are relatively coherent but parts of the figures are difficult to see and the captions are difficult to understand. I have reviewed an earlier version of this paper. Although the authors have revised the manuscript to take into account some of my earlier suggestions, I was quite disappointed that many of them were simply ignored. I repeat some of them here. I strongly encourage you to adjust your manuscript and resubmit. The temperature proxy community needs to be made aware of the inconsistencies in assuming constant isotope ratios.

This is especially important in light of the application of trends from palaeotemperature distributions to current climate change models.

We thank the Reviewer #2 for his/her support.

15. You present two scans, one before and one after treatment but I could not find where you tell how many experiments you did or why you are convinced that our conceptual model of past climate should be revised based on this one experiment. I believe you because your results are consistent with what I have seen and what other colleagues have discussed with me over the past 20 years but you need to convince the isotope temperature proxy community. What is your uncertainty? Reproducibility? I know that Nano-SIMS time is expensive and it is not easy to convince those who control access, of the necessity for reproducibility experiments but the manuscript lacks information about just how solid those very nice images are.

We have added more details about NanoSIMS in the section Methods. Note that the recent review paper by Hoppe et al. (2013), cited in our manuscript, provides full (but readable) technical details on NanoSIMS, illustrated with many examples of applications. If requested, we can repeat more of this information in the section Methods of our manuscript (cf lines 222-237).

The NanoSIMS is an instrument primarily designed to analyze large chemical or isotopic differences (so that high precision is not a requirement), in situations when high spatial resolution is needed. This is precisely the case here.

The NanoSIMS is capable of quantitatively imaging isotopic distributions with a lateral resolution of around 100 nm, by bombarding the surface of a sample with a primary beam of ions (here Cs⁺) that sputters off atoms and small molecules, some of which are ionized (the secondary beam) and can be analyzed in a multi-collection mass spectrometer using electron-multiplier detectors.

Now, producing a primary beam with a spot size of ca. 100 nm has a price, which is that this beam is very low intensity, i.e. relatively few Cs⁺ ions hit the sample surface per second (compared to conventional ion microprobes). This has the effect that the number of secondary ions generated from the sample is correspondingly low, and isotopic ratios obtained with the NanoSIMS from a given 100 nm-sized pixel are low precision, typically a few %.

Here, pure ¹⁸O has diffused into foraminifera tests, leading to isotopically heterogeneous tests with areas highly enriched compared to others (factors of almost 1000). In such a case, we do not need high precision. This heterogeneity exists at a very small scale, which the NanoSIMS can resolve. In other words, this type of imaging is precisely what the NanoSIMS is constructed to do.

Of note: Although the precision is low, the reproducibility is high (we obtained the same results [within uncertainties] for all the measures we made on triplicate samples coming from different runs).

16. One cannot "simulate" diagenesis. Saying it this way is quite offensive to geologists and simply incorrect. Please use other words that objectively describe what you did. You increased temperature and pressure, in an attempt to hasten recrystallisation or isotope exchange. One cannot simulate a process, that takes millions of years, in 3 months or 30 years.

It is true that we increased temperature and pressure, in an attempt to hasten isotope exchange, thereby simulating diagenesis. We agree that we cannot “reproduce” diagenesis in the lab, but we can simulate it. Simulating actually means “making in imitation of”. In any case, we now avoid using the words ‘simulated diagenesis’ in the manuscript.

17. We are missing basic information about how Nano-SIMS works so we can understand the results. Tell us briefly what advantages and limitations it has for your samples. Tell us if it measures in the top few molecular layers, i.e. the surface, or it measures the whole section that you cut out from the SEM image. What are the artefacts of the technique? The general reader needs to know this in order to appreciate the information provided by the images. This information could be put into the section in Methods where you provide the instrument details.

See answer to point 15.

18. In the main text: "added in order to reach the ionic strength and alkalinity of seawater and limit the extent of dissolution/precipitation reactions that might otherwise occur during the quench phases of the experiments" This does not make sense.

We removed this sentence.

For the record, if we had used pure deionized water, foraminifera would have dissolved until reaching saturation with respect to calcite at room temperature. Worse, the solubility of calcite being a decreasing function of temperature, calcite would have re-precipitated at high T and calcite would have re-dissolved again at the end of the experiments during the quenching phase. All this dissolution/precipitation/re-dissolution sequence would have modified the results of our experiments.

Increasing alkalinity of our solution allowed minimizing this issue. In fact, as stated in the Methods section, with the solution we used, at ambient temperature (293 K), assuming thermodynamically controlled dissolution of calcite (i.e. neglecting kinetic barriers), equilibrium was reached after dissolution of less than 2% of the foraminifera tests. At the experimental temperature (573 K), because of the retrograde solubility of calcite, saturation with respect to calcite was reached for a dissolution progress corresponding to only 0.1% of the tests. Here, NanoSIMS imaging revealed that the average $^{18}\text{O}/^{16}\text{O}$ ratio of the foraminifera tests, initially about 0.002 (Fig. 1), became as high as 0.15 during experiments (Fig. 2), equivalent to the replacement of about 15 vol% of the initial biogenic calcite by pure $\text{CaC}^{18}\text{O}_3$. Therefore, we can assume that the NanoSIMS images reflect isotopic re-equilibration very close to chemical equilibrium, with little or no contribution from secondary crystallization from the bulk solution. We made this point clearer in the revised version of our manuscript (cf lines 198-209).

19. In the methods section: "In order to limit the extent of dissolution/precipitation/re-dissolution reactions that could occur during the experiments as a consequence of differences in calcite solubility..." It is not clear to me how dissolution/precipitation would be minimised during change of temperature by increasing ionic strength and alkalinity. It is the ion activity product that defines solubility. I suggest you simply report the solution concentrations and leave out this explanation. Adding salt actually increases solubility and the rate of recrystallisation.

We removed this sentence (we meant ‘compared to using pure deionized water’).

20. "To avoid artifacts possibly created by complicated, multi-step cleaning procedures, the foraminifera tests were..." Why do you write this if you did not do any?

We removed this sentence.

21. You used "pure ethanol" in an ultrasonic bath to clean the samples. What needed to be cleaned away? Why is this important? What is "pure" ethanol? Standard pure ethanol is usually only about 96% with the balance water. This is plenty of water for dissolving calcite, especially if it is in an ultrasonic bath, through 3 rinses. Calcite could dissolve along grain boundaries, increasing surface area for adsorption and surface exchange of ^{18}O , which would have an effect on the parameters you use in the model. Even if the purity is 99% as a result of distillation, exposure to air brings the water proportion back to 96%. Also, exposure to ethanol results in a very strongly bound surface layer which is not displaced easily by water. Can you be sure that the ^{18}O that is removed from solution is not associated with the adsorbed ethanol? What happens if you make your experiments without cleaning the forams ?

The foraminifera that we used for our experiments are sub-modern natural foraminifera that come from modern sediments. These foraminifera are covered by clays that have the ability to absorb a non-negligible amount of water. Results would have been distorted if experiments were performed without removing these clays prior to the experiments. Cleaning foraminifera at the end of the experiments was also necessary to remove the ^{18}O -pure water that may adhere to the surface of the tests at the end of the experiments. We made this point clearer in the revised version of our manuscript (cf lines 76-78). In any case, it should be kept in mind that we documented here ^{18}O diffusion within foraminifera tests (see Figure 2), not surface enrichments.

22. "Activity coefficients for aqueous species were calculated using the Davies equation, which remains satisfactory for ionic strengths of up to 0.6 mol/L." This is close to the ionic strength of sea water where the Davies equation certainly does not hold, especially for divalent ions. This is stated in numerous textbooks on aqueous speciation (Stumm and Morgan, Garrels and Christ and many others) so citing one paper where the authors used Davies is not good enough as a base for such strong claims as your paper needs to make. But I do not think that your inability to predict saturation state absolutely precisely for your solutions is a show stopper. Whether or not they have completely reached thermodynamic equilibrium, or even isotopic equilibrium, only adds some uncertainty to your time and temperature estimates – which is ok. You cannot claim valid nonideality corrections for sea water conditions using Davies. That is why Pitzer worked so hard to find an alternative. Even the old mean salt method is better than Davies for sea water. So instead, just write what you did and leave it at that. "Activity coefficients for aqueous species were calculated using the Davies equation."

We modified the sentence according to the Reviewer's comment (cf lines 199-200). For the record, we agree that the Pitzer model appears slightly closer to experimental data for ionic strengths of about 0.6 mol/L compared to the Davies one (see the figure below from Samson et al., 1999 for instance). We used the Davies equation because the parameters required to use the Pitzer equation do not exist for high T conditions.

Comparison of NaOH activity coefficients for different models (Source: Samson et al., Comp. Mat. Sci., 1999)

23. "Simple calculations show that, with this process running to completion, the isotopic signatures of fossilized benthic foraminifera would be completely equilibrated in less than 10^5 years, regardless of the temperature conditions." Rate of exchange depends on how far the system is from equilibrium so extrapolation is not simple. The isotope ratios in your experiments are far from equilibrium so the extent of change you see is clear – and you want it to be, to make your point. But the pore fluids during diagenesis are close to equilibrium. I am not convinced that you have made the assumptions clear enough.

This sentence is absent from the manuscript that we submitted for publication in Nature Communications. Anyway, we agree that the distance from equilibrium constitutes a driving force for re-equilibration when dissolution and precipitation reactions are considered. On the contrary, if solid-state volume and grain boundary diffusion is considered (as we do in our study), the characteristic diffusion length will only depend on the diffusion coefficient of the considered species (O) in the considered medium (calcite). We made this point clearer in the revised version of our manuscript (cf lines 111-128 and the Methods section).

24. You use the words "interfacial/pseudomorphic dissolution-precipitation" many times but you have not used the terms correctly and your explanation is not clear. What is "interfacial" dissolution? Dissolution between faces? What is this? In general, throughout the paper, you make adjectives out of words that should remain nouns. In this case, I think you mean dissolution and precipitation at calcite interfaces where the crystal structure is not altered. So say that. A pseudomorph is a crystal that grows within the volume that was previously occupied by a crystal with a different composition and sometimes, different symmetry. Calcite replacing calcite is not a pseudomorph according to the definition used by mineralogists. It is important that the words you use are correct in the scientific communities who recognise the topic as belonging to them or your results are not respected. You can simplify the explanation to make it more clear and correct. I suggest the following changes: "Mechanistically, the observed oxygen isotope exchange could have occurred either through dissolution and precipitation at interfaces between calcite crystals and the solution, which would exchange the calcite while it retained its original shape^{22,23}, or through much slower replacement by movement along grain boundaries or by solid state diffusion, i.e. the movement of oxygen atoms within the biogenic calcite matrix without dissolution²⁴⁻²⁶."

We removed this terminology. We now refer to "coupled dissolution and precipitation at mineral–fluid interfaces" as defined by Ruiz-Agudo et al., Chem Geol, 2014).

25. "Dissolution and reprecipitation at interfaces requires a difference in molar volume between the primary and secondary phases to generate porosity that would preserve contact between the aqueous phase and the dissolving solid²³. This condition is not met in foraminifera tests, because the primary and secondary phases are both calcite." This statement does not make sense. If you remove an ion one place and put it somewhere else, there is no volume change, and volume change is not necessary, but the interface between solid and solution remains. Of course you would have both processes in your exchange experiments.

This is true. But because the exchange reaction in our experiments was not limited to the first monolayer of O atoms bonded to calcite (Figure 2), it resulted from either (1) coupled dissolution and precipitation at mineral–fluid interfaces (e.g. Ruiz-Agudo et al., Chemical Geology, 2014; Putnis, Science, 2014) or (2) solid-state volume and grain boundary diffusion (e.g. Farver and Yund, EPSL, 1998; Stipp, Nature, 1998).

As emphasized in Ruiz-Agudo et al. (2014): “In order to propagate a replacement front, mass transfer pathways must be maintained between the fluid reservoir and the reaction interface. This requires that the replacement process is a volume deficit reaction, and that the resulting product is porous (...) and hence allows continued infiltration of the fluid phase to the interface with the parent phase. This porosity results from both the molar volume differences between parent and product as well as the relative solubilities of the phases in the specific fluid at the interface”.

Because there is no volume deficit reaction in the replacement of ¹⁸O-poor calcite by ¹⁸O-rich calcite, we concluded that the first mechanism does not apply here and thus used the (much slower) second mechanism for our numerical simulations, thereby exploring the most “conservative” scenario with respect to the preservation of the original isotope signal.

We made this point clearer in the revised version of our manuscript (cf lines 97-110).

26. For Figure 1, can you clarify the last 3 or 4 lines of the caption? It took me several readings to understand what was what. We need scale bars we can read. "Both carbonates and clays exhibit a normal 18O/16O ratio of 2.10-3." It might be smart to write 0.002 to make this number more easily comparable to the numbers in the caption to Figure 2.

We made the figure caption clearer and modified the figure accordingly.

27. For Figure 2, we need scale bars we can read.

Done.

28. The caption for Figure 3 needs to be rewritten so we can understand what the figure shows. The labels on the figures need to be made so they are readable.

We made the figure caption clearer and modified the figure accordingly.

29. The caption for Figure 4 is altogether too long and certainly not clear. Can some of this be put into the text? The text on the figures is too small to be readable.

We made the figure caption clearer and modified the figure accordingly.

30. I did not review the sections in Methods about re-equilibration and the model. Frequently, you write "cf XXX". Explain things logically, then you don't need to send us off to other places in the paper to find the information we need. It is disruptive and loses the reader's attention.

We modified these sentences accordingly.

31. A number of suggestions:

- "ultra" is a prefix meaning extreme. "Ultrastructure" has no meaning. How about microstructure?

The word ultrastructure has a meaning in paleontology. It stands for the nanostructure of a specimen at a small scale.

- In the abstract, "Numerical modelling showed..." Numerical modelling of what?

Numerical modelling of diffusion during burial. We made this point clearer in the revised version of our manuscript (cf line 29).

- "However, subsequent to the recognition of bias from secondary calcite crystallization in early data..." You can say "secondary calcite formation" or "calcite recrystallisation" (which is what I think you mean) but "secondary calcite crystallisation" does not work.

We modified this sentence (cf lines 47-48).

- "Such low thermal gradients are not possible to reconcile with climate and ocean..." How about: "...are not consistent with..."? (This is a very nice problem statement.)

We modified this sentence (cf lines 53-54).

- " An ultra-high resolution isotope imaging technique, such as NanoSIMS, seems required to visualize the isotopic resetting of foraminifera tests occurring during diagenesis in oceanic sediments." This sentence does not fit the flow of the paragraph. Isn't this more what you mean? "An ultrahigh resolution imaging technique, that can detect small changes on relatively short time scales, would help solve this problem. Nano-SIMS is such a technique. It uses... (short explanation of how it works and then put info about the advantages and limitations and common artefacts in Methods). "

We modified this sentence according to the reviewer's suggestion and added details on NanoSIMS in the Methods section.

- "raising questions about the reliability of fossil foraminifera..." Do you trust your results enough to write: "raising serious questions"

Yes we do. We thus modified this sentence accordingly (cf lines 89-93).

- " ...compare the slopes of curves in Fig. 4a." How about: "such as is shown by the slopes..."

We modified this sentence according to the reviewer's suggestion (cf lines 158-161).

- "Paleotemperatures inferred from" – "Paleotemperatures interpreted from".

We modified this sentence according to the reviewer's suggestion (cf lines 162-163).

- Figure 3, "Sediment temperatures, corresponding to typical geothermal gradients between 40 to 60 °C.km⁻¹ and deep seawater temperature of 3.5 °C, are indicated. " Indicated where?

Indicated on the Figure; we made this point clearer in the revised version of our manuscript (cf Figure Legends).

- "100 µL of this synthetic 'seawater' solution" No need for quotation marks – and these are apostrophes.

OK.

- "the mass balance is such that the water δ18O can be considered constant during the experiments, regardless of the magnitude of the isotopic exchange during the experiments." Please tell us the mass of solid to solution ratio. This is a more telling measure of how likely you are to deplete the solution reservoir.

About 160 µg of cleaned foraminifera tests were immersed into 100 µL of the synthetic seawater solution. Therefore, 1.6 g.L⁻¹ foraminifera have been immersed within a solution consisting of pure H₂¹⁸O water containing NaCl (0.55 mol.L⁻¹) and NaHCO₃ (0.003 mol.L⁻¹). Since the ¹⁸O/(¹⁶O+¹⁸O) ratio of foraminifera is close to 0 compared to that of the solution (which is 1), the ¹⁸O/(¹⁶O+¹⁸O) ratio of the total system {solution + foraminifera} was equal to 0.998.

- "the solution is slightly undersaturated with respect to calcite." – within uncertainty, right?

Yes, we modified the sentence accordingly (cf lines 201-202).

- "Therefore, we can confidently assert that the NanoSIMS images reflect isotopic re-equilibration very close to chemical equilibrium, with little or no contribution from secondary crystallization from the bulk solution." This is a strong and good statement but replace "assert" with "assume" – and leave out "confidently". There is some uncertainty in your experiments as there always is, so "assume" is good enough. It doesn't matter because Ostwald ripening guarantees some dissolution and recrystallisation, which is what you are after in the isotope re-equilibration anyway.

We modified this sentence accordingly (cf lines 207-209).

- Carnegie Institute, not Institution?

No. This is Carnegie Institution of Washington.

32. The paper is in need of editing by someone who understands English grammar.

There are far too many examples of nouns converted to adjectives unnecessarily. I pointed this out in an earlier review and was disappointed to see as many, if not more, grammatical errors: hydrospherE evolution ... oxygen isotopE composition ... foraminiferA tests

geologic timescales ... climate signals... foraminifera calcite tests ... the O isotope signal ... subsequent to (no ly) and many more (where the upper case letter shows where the word should end and not have an adjective ending added). There are many examples of carelessness, where standards should be followed. Species names should be written in italics. "May" is used only for permission; "could, can, would" are used for showing possibility. "As" should never be used as a synonym for "because". "As" is a preposition, "Because" is a conjunction. There are many unnecessary hyphens. Past tense should be used for things you did and things you observed. Present tense is used for things that always happen. Future tense should not be used for things that always happen (i.e. "... change in fossil test bulk O isotopic composition would, if isotopic resetting is ignored, ...". "Due" is for library books and bills to pay; it should not be used to replace more explanatory words such as because of, from, in response to, and about 46 other possibilities. Empty words can be removed, such as "for the present study", "respectively", "see" and many others. In fact, in this paper, "respectively" has been used incorrectly and unnecessarily 6 times. Simple rearrangement of the sentence uses fewer words and is less ambiguous. IUPAC standards need to be adopted, Ma not Myr, space between the unit and the quantity, also for °C, i.e. 25 °C.

We edited the manuscript following the Reviewer's remarks and we commit to enlist the Nature Research Editing Service to assist us with English grammar if our manuscript is accepted for publication.

Reviewer #3

33. Bernard et al. designed a fascinating experiment with results that shouldn't be ignored in future studies which are based on the oxygen isotope composition of Cretaceous and Early Paleogene foraminifera shells: Most scientists share the view that fossil foraminifera shells featuring no visible signs of recrystallization or cementation largely retain their original $\delta^{18}\text{O}$ composition (this is less certain for their minor and trace element composition). However, Bernard et al. demonstrated that the original $\delta^{18}\text{O}$ in shells that show no visible signs of alteration is not necessarily preserved and thus introduced a new potential challenge to paleoceanographers. In a well-designed experiment, Bernard et al. sealed modern shells of *G. bulloides* into gold capsules with 'heavy' water (H_2^{18}O), sodium bicarbonate, and sodium chloride, and exposed them at 300°C under a pressure of 200 bars (~2000 m) for 3 months. A control experiment was performed with isotopically normal water. Subsequent $\delta^{18}\text{O}$ analyses by NanoSIMS revealed that isotopic exchange occurred within the shells exposed to isotopically heavy water, leading to a significant enrichment in ^{18}O , whereas – as expected – the shells from the control experiment retained their original $\delta^{18}\text{O}$ (within analytical uncertainty of the NanoSIMS). SEM images document that the change in $\delta^{18}\text{O}$ is not accompanied by modifications in shell structure.

We thank the Reviewer #3 for this positive summary.

34. The first part of the manuscript was a pleasure to read, precise, and well-written, and I wonder why no one performed this important experiment before. This experimental design and the results clearly warrant publication in Nature Communications, however, I am not entirely happy with the second part of the manuscript, in particular with the interpretation of the results. Thus, I recommend moderate revisions.

Below, I list my main concerns, and hope the authors won't regard this as criticism but as an opportunity to further improve the manuscript.

We thank the Reviewer #3 for his/her support.

Please find below our detailed answers to the Reviewer #3's main concerns.

35. The authors compiled several models demonstrating the potential impact of this “diagenetic diffusion controlled O-isotope equilibration” on the veracity of benthic $\delta^{18}\text{O}$ records over the timespan of 120 million years (Fig. 3), or they demonstrate that the low equator-to-pole temperature gradient during the Paleogene, which is plaguing climate modelers for decades, may be resolved when the diffusion controlled oxygen isotope modification is taken into consideration. In this respect, it should be mentioned that several published papers already discussed the potential effect of diagenesis on the meridional temperature gradient deduced from planktic $\delta^{18}\text{O}$, such, this model and concept is not new, only the mechanism applied to explain the modification of foraminiferal $\delta^{18}\text{O}$ is different.

This is true that several authors already discussed the potential impact of secondary crystallization with the conclusion that recrystallized foraminifera should not be used for paleoclimate reconstructions (see answer to point 6). Here, we investigated the impact of diffusion, a process that leaves no visible trace, in contrast to recrystallization. We concluded that, above a certain age, foraminifera that do not look recrystallized, a.k.a. the glassy ones, are likely to have had their O isotope composition altered through diffusion. This is more than simply a matter of mechanism. We made this point clearer in the revised version of our manuscript (cf lines 89-93).

36. These models are certainly valid and interesting - from a modeling point of view – but they are somehow disconnected from observations and published data and as such of limited value for the paleoclimate community.

We are surprised by this comment given that we used, for our numerical simulations, real values from the DSDP/ODP/IODP reports for the sites used by Zachos et al. and Friedrich et al. to build their compilations (see answer to point 5). We made this point clearer in the revised version of our manuscript (cf lines 137-141 and 453-471).

37. There is one missing link: The diagenetic diffusion controlled O-isotope equilibration would be a huge challenge IF fossil shells would appear pristine – because one would not expect a diagenetic bias in $\delta^{18}\text{O}$. However, shells having reached the critical age for this process to take place (~ 30 Ma, lines 181-182), are typically visibly affected by recrystallization or cementation. With the discovery of well preserved ‘glassy’ shells and the landmark study by Pearson et al., (2001), demonstrating that the $\delta^{18}\text{O}$ of ‘glassy’ planktic shells is significantly lower than that of their ‘frosty’ counterparts, paleoceanographers are now aware of the potential diagenetic bias, and most scientist would not take the $\delta^{18}\text{O}$ composition of >30 Ma old shells at ‘face-value’. The potential bias in the $\delta^{18}\text{O}$ of fossil planktic and benthic foraminifera is now well known and, although difficult to quantify. Therefore, the relevant findings can be summarized to the discovery of an additional (diagenetic) mechanism, that can, in concert with cementation and recrystallization, modify the original shell $\delta^{18}\text{O}$. Thus, this discovery may be less alarming than it may sound in the beginning.

We agree that the $\delta^{18}\text{O}$ composition of 'frosty' old foraminifera tests certainly should not be trusted to record paleoclimate temperatures accurately (but note that many paleoceanographers (wrongly) consider that recrystallization or cementation occurred prior to burial and thus do not impact the O isotope composition of benthic foraminifera). However, our study shows that even 'glassy' foraminifera O isotope compositions have likely been altered during burial without any visible trace of alteration. We made this point clearer in the revised version of our manuscript (cf lines 89-93).

38. Also, the title of the manuscript can be modified, as diagenesis is the main driver. O-isotope re-equilibration may be a contributing factor.

The re-equilibration is the process that occurs, not a contributing factor, and it occurs during burial. The driving force for re-equilibration of foraminifera tests through diffusion is the disequilibrium between the tests and the surrounding pore water. This disequilibrium increases with increasing temperature, i.e. with increasing sediment burial (cf lines 111-116).

39. Statements like "Cretaceous and Paleogene deep oceans were likely much colder than previously thought" (lines 163-164) can certainly be made (also I would say "potentially" instead of "likely"), however, this is due to the fact that many older studies did not consider potential diagenetic alteration, thus, the shells were not screened carefully prior to analyses.

We did not investigate the impact of secondary crystallization. We investigated the impact of diffusion and demonstrated that even 'glassy' foraminifera O isotope compositions have likely been altered during burial without any visible trace of alteration (cf lines 89-93). Carefully screening fossil foraminifera prior to analyses will not allow avoiding this issue. Corrected for burial induced isotope re-equilibration, a low temperature for Cretaceous and Paleogene deep oceans is re-established in the foraminifera O isotope record.

40. This is particularly true for benthic foraminifera as they are considered as relatively resistant to diagenetic alteration compared to their planktic counterparts (although this view is somehow changing). In this respect, the authors should discuss if the results of their experiment can be unambiguously applied to benthic shells. Bernard et al. performed their experiment on planktic shells (*G. bulloides*). Compared to benthic shells, the surface/mass ratio of planktics is much larger. Will the results be the same?

Because the carbonate 'building blocks' of benthic and planktonic foraminifera are the same at the microscale (Cuif et al., 2011), the process we described and quantified impacts similarly both planktic and benthic foraminifera. We made this point clearer in the revised version of our manuscript (cf lines 210-213). See also answer to point 6 above.

41. However, I don't want to downplay the fascinating results from the experimental design by Bernard et al. But I feel that there are other important questions that should be addressed and discussed: For example, is it possible that the $\delta^{18}\text{O}$ of 'glassy' foraminifera shells, that are currently regarded as a 'gold-standard' for the accurate reconstruction of Cretaceous and Paleogene ocean temperatures, are biased by O-isotope equilibration that leaves no visible changes in shell-structure?

The answer is yes, this is exactly the conclusion of our study, with all the inherent implications.

42. Some other aspects the authors may consider:

□ while it is clear that models do not necessarily need to reflect ‘real world’ observations, some items could be improved: In particular, I have some problems with Fig. 4b. The authors take the modern latitudinal temperature profile and project it back 100 Ma using the effect of simulated diagenesis on $\delta^{18}\text{O}$, and combine this with Paleogene $\delta^{18}\text{O}$ data from planktic foraminifera for comparison. However, the climate conditions in the Paleogene, when these foraminifera lived, were very different from those today, and cannot be modeled by a diagenetic backcalculation of the modern temperature gradient. In other words: The authors imply that the latitudinal temperature gradient was within the past 100 Ma similar to the gradient we observe today, and the apparent ‘weaker’ gradient during Paleogene and late Cretaceous climate is mainly a result of diagenesis (line 227 ff.). This is definitely too simple.

Our model allowed simulating changes in the foraminifera O isotope signal along a latitudinal temperature profile similar to the modern one after a burial of e.g. 45 millions of years. Results are unambiguous: the effect O isotope re-equilibration during sediment burial is to significantly flatten the inferred paleo-latitudinal temperature gradient on such a time scale (cf Figure 4b).

Corrected for burial-induced isotope re-equilibration, a steeper temperature gradient between low and high-latitude surface ocean waters is re-established for the Paleogene. Such a gradient resolves the paradox of the Paleogene low equator-to-pole surface ocean thermal gradient and is consistent with climate and ocean circulation models.

□ the authors use the $\delta^{18}\text{O}$ of pore water in combination with burial temperature to calculate the modification of shell $\delta^{18}\text{O}$ by diffusion controlled O-isotope equilibration over time. However, porewater profiles are very complex, and the $\delta^{18}\text{O}$ of pore water does not necessarily reflect the $\delta^{18}\text{O}$ of the water when the foraminifera calcified (e.g. Paull et al., 1995).

We agree that the O isotope composition of the sediment pore water in which foraminifera were buried can have locally changed. We deliberately assumed that the pore water kept the same O isotope composition as the seawater in which foraminifera lived (i.e. -1‰) following Schrag et al. (Science, 1996). This is confirmed by most of the DSDP/ODP/IODP reports that show that the $\delta^{18}\text{O}$ of the pore water in old sediments is close to -1‰ (this supports the estimation of the $\delta^{18}\text{O}$ of the ice-free ocean). We made this point clearer in the revised version of our manuscript (cf lines 135-137 and 439-451).

□ Furthermore, it is not known if the concept of pore fluids is applicable to model calculations of diagenesis. It was suggested that dissolution and recrystallization must be taking place in very small cavities or aqueous films within the solid test wall, and these solutions may have a very different chemical composition than the true pore fluids that circulate between the tests and within the chambers (Pingitore, 1982; Pearson & Burgess, 2008). I feel that these aspects need to be discussed.

We agree that small cavities may have a very different chemical composition and may impact dissolution and recrystallization processes. In contrast, if solid-state volume and grain boundary diffusion is considered (as we do in our study), the characteristic diffusion length will only depend on the diffusion coefficient of the considered species (O) in the considered medium (calcite), no matter the chemical composition (i.e. saturation state with

respect to calcite) of the solution. We made this point clearer in the revised version of our manuscript (cf lines 122-128 and the Methods section).

43. Minor points to address:

line 68: how many shells are approximately 150 μg ?

150 μg of foraminifera correspond to 12 of them.

line 220: are these paleolatitudes? They differ from the values given in the publications

Yes, these are paleolatitudes. We made this point clearer in the revised version of our manuscript (cf Figure Legends).

Reviewer #4 (Remarks to the Author):

See attachment

Reviewer #5 (Remarks to the Author):

Review of "Burial-induced oxygen isotope re-equilibrium of fossil foraminifera explains ocean paleotemperature paradox."

The authors has succeed to make significantly important evaluations with new hypothesis on global trend of stable oxygen isotopic profile and there is no doubt that it will have a big impact on the audience. Because the $\delta^{18}\text{O}$ is quite basic tool for broad field of earth science, the influence of this study is not limited to paleoceanography, it is considered to be an important knowledge on the perspective of earth evolution for whole Cenozoic. This study give a clear answer to the hypothesis. The authors show a very good approach to solve this question with appropriate fancy methods of measurement and numerical modeling. The reviewer can identify the study is suitable for publishing in *Nature Communication*.

Major Questions

Have authors attempted oxygen isotopic mapping using foraminifera of the geological era (for example, from 40 Ma, 50 Ma)? If authors have already tried to visualize them, describe/indicate the result, regardless of success or failure. On the other hand, the reviewer imagines that such a measurement is not attempted. Perhaps the reviewer thinks that the effect of re-equilibrium could not be visualized by the current analytical accuracy of oxygen isotope ratio by Nano-SIMS. Therefore, the reviewer speculates that it is necessary to conduct this simulation experiment of re-equilibrium in laboratory and computer.

Even so, the author have to specify the reason why they does not show directly re-equilibrium imaging in the geological samples, in order to explain to the reader why the design of the experiment has become the current strategy.

It is also necessary to explain about the reliability of the model has been guaranteed. The constructed diffusion model is the foundation in this study, but I could not find a description of how this correctness was evaluated. This is a just example, if authors enter the parameter of the present experiment in the numerical model, can they recursively check whether the model itself and the measured value are within same range or not, and whether reliable results can be obtained?

Related to this, it is necessary to specify whether the crystal size obtained by equation 15 agrees with the known size by previous studies (50-200 nm, L119). This is also helpful in evaluating the correctness of the model. Though the authors has described "the mean calcite size can be calculated by solving for r_0 in the following equation.", but the reviewer can not find the calculation result.

Does the same influence of re-equilibrium cab be appeared in the carbon isotope ratio as well as the oxygen isotope ratio?

L45: relatively cold tropical surface...

I guess it is also newly explained in this study. Authors can mention about this wrong previous estimation in discussion as well as warmer estimation at higher latitude.

L119: Iwasaki et al. (2015, *Paleoceanography*: 10.1002/2014PA002639) also show clear image of calcite particles of planktonic foraminifera (*G. bulloides*). The reviewer thought this would be nice to see for authors.

L191: The species name would be Italicize.

L566 The spelling will be fixed. 31. van der Lee, J. & De Windt, L. CHESS Tutorial and Cookbook.
U"p"dated for Version 3.0.

Reviewer #6 (Remarks to the Author):

See attachment

Revision of “Burial-induced oxygen isotope re-equilibration of fossil foraminifera explains ocean paleotemperature paradoxes” by Bernard, Daval, Ackerer, Pont, and Meibom.

Response to Referees

Reviewer #3

2nd Review: “Burial-induced oxygen isotope re-equilibration of fossil foraminifera explains ocean paleotemperature paradoxes” submitted to *Nature Communications*

This is my second review of the manuscript that Bernard et al. resubmitted after minor to moderate revisions. The authors addressed most of my comments (and those of the other reviewers) in a – mostly – satisfactory way; still, I am hesitant to fully recommend this manuscript for publication in its current form, mainly due to two issues:

1) The authors simulate diagenesis at realistic pressures that can also be found in deep sea sediments; however, in order to accelerate the experiment (which they need to do), the authors increased the reaction temperature to 300°C (for 3 months). These temperatures are significantly higher than those observed in sea floor sediments from burial depths used to recover foraminiferal shells for paleoclimate reconstructions (typically, buried foraminiferal shells are exposed to burial temperatures well below 20°C. This fact is also acknowledged by the authors, see line 143). I don’t know if anyone can say with certainty **if the increased experimental temperature is just accelerating the process of O-isotope re-equilibration, or if the O-isotope re-equilibration, as described by Bernard et al., only occurs as a result of these P-T conditions. Even if the geochemical modelling looks solid, the actual chemical processes in the sediment column that occur over time spans of 10s of millions of years may be much more complex and comprise additional parameters/factors that cannot be considered in current models or experiments.** Thus, I believe that O-isotope re-equilibration is a **possible** scenario that **may** affect (in concert with diagenesis) the $\delta^{18}\text{O}$ of fossil foraminiferal shells after 10s of millions of years. Having this said, I would prefer a more careful wording. I don’t feel comfortable that the authors have written and worded the manuscript in a way that implies that their findings from the O-isotope re-equilibration experiments can be unambiguously applied to the natural environment (e.g. line 26: O-isotope re-equilibrium of fossils foraminifera tests cause; line 156: the benthic foraminifera O-isotope record primarily reflects burial-induced isotope resetting..., line 158: deep oceans were likely much cooler..., line 181: this process has likely attenuated...). I encourage the authors to ‘soften’ their statements (e.g., has the potential to cause..., could possibly reflect..., could have been much colder..., has potentially attenuated...), and so on.

We softened some statements accordingly.

2) Good science and the quality of writing goes hand in hand. I strongly encourage the authors to ask a proficient English speaker to go through this manuscript. I believe that publications in a high-impact journal such as *Nature Communications* should also be exceptionally well written. Clearly, there is still some room for improvements.

We edited the manuscript and we commit to enlist the Nature Research Editing Service to assist us with English grammar if our manuscript is accepted for publication.

Nevertheless, I think (in agreement to my first review of this study) that this manuscript – after some mild revisions – warrants publication in *Nature Communications*. The findings of Bernard et al. will foster important discussions in the paleoclimate community, although not everyone will agree with their interpretations, and likely promote the design of follow-up experiments.

We thank the Reviewer for his/her support.

Some minor suggestions:

- line 25: scanning electron scanning microscopy

Done

- line 27: sub-micrometer length scale

Done

- line 27 length

Done

- line 29 'has the potential to cause

Done

- line 32 may have been (instead of 'were')

Done

- line 46 cool tropics paradox (tropics typically plural)

Done

- line 48 also known as (instead of a.k.a.)

Done

- line 49 significantly (instead of slightly. The difference in paleotemperature estimates from frosty and glassy tropical planktic shells is significant)

Done

- line 54 delete “which requires explanation”. It should be mentioned that climate models are also far from being perfect (and climate modelers acknowledge this). Thus, it is not fair to say that the disagreement between proxy data and model simulations is only based on bad proxy data.

This is true. We deleted this part of the sentence.

- line 61 “easily” is the wrong word. It is impossible to reproduce burial diagenesis in the lab without accelerating the process

Done

- line 118 Scanning electron

This sentence refers to scanning and transmission electron microscopy.

- line 147: a seawater

Done

- line 165: effect of

Done

- line 170, 171: climate models are not perfect either

This is true (cf comment above).

- line 173 reword “removed the requirement”. This is not a requirement. It is the current interpretation of the benthic $\delta^{18}\text{O}$ stack.

Done

- line 181: processes have

Done

- lines 214: write: we propose that the results of this experiment, although conducted on planktonic foraminifera, can also be applied to benthic species. (mentioning paleotemperatures is already an interpretation of the results – first, the authors need to say that the results apply to both planktic and benthic forms)

Done

- line 226: ion microprobe

Done

- line 228: high mass resolving power

Done

- line 229: $^{133}\text{Cs}^+$ ions

Done

- line 234: I thought the beam is rastering (not stepping)

Done

Reviewer #4

Review on Bernard et al. 'Burial-induced oxygen isotope re-equilibration of fossil foraminifera explains ocean paleotemperature paradoxes'

The authors have investigated a mechanism (bulk diffusion of oxygen in calcite) that could be responsible for the change of a proxy signal (^{18}O) over long (> 10 Ma) time scales. Such a slow process cannot be observed directly, not even in experiments that would last the timespan of a long career in science. Anyhow, Bernard et al. found support for this mechanism by an experiment that lasted 3 months and under modified conditions (higher temperatures) based on solutions of the diffusion equation using appropriate parameterisations for the diffusion coefficient. Their investigations suggest that the ^{18}O composition in calcite precipitated by foraminifera and archived in marine sediments can have been modified by a large amount (several ‰) over long time scales. If true, these findings would request a large change of our reconstructions of the marine temperature development over timescales > 10 Ma with consequences for our understanding of the functioning of Earth system including climate.

We thank the Reviewer for this positive summary.

Many aspects of the paper have been already discussed between the authors and three reviewers. I was impressed by the high level of this conversation.

We thank the Reviewer for this positive comment.

Although bulk diffusion of oxygen in calcite and consequences for changing of ^{18}O distributions in calcite are scientifically sound and have been demonstrated convincingly, the uncertainties are still large. This includes the quantification of the diffusion coefficient (commonly described by an Arrhenius type equation depending on a coefficient D_0 in front of the exponential function and the activation energy E_a). However, the size and composition of the calcite crystals could also play a role.

See discussion below regarding D_0 .

Despite these uncertainties, I recommend publication in Nature Communications because of the high quality of the investigation and the potential importance of the results. Publication of the manuscript will stimulate further investigations that will hopefully reduce some of the uncertainties mentioned above. The results could help to resolve some of the paradoxes mentioned in the paper as well as in the discussions between reviewers and authors. In a few years, the article could probably be seen as a seminal paper at the begin of a paradigm shift.

We thank the Reviewer for his/her support.

Major comments/suggestions: Discussion of the term:

$$D_0 \exp\left(-\frac{E_{a,foram}}{R T_{xp}}\right) t_{xp} \quad (1)$$

in Eq. (6) would help the reader to understand the experimental approach.

In order to obtain a change in isotopic composition this term has to be of a certain size which can be achieved by either long time (t , in sediments) or large temperatures (T , in lab). Let us equate the term in Eq. 1 for two combinations of temperature T and time t :

$$D_0 \exp\left(-\frac{E_a}{R T_1}\right) t_1 = D_0 \exp\left(-\frac{E_a}{R T_2}\right) t_2 \quad (2)$$

which for the chosen values yields $t_2 \approx 10$ Ma, a value given, however, not derived in the manuscript (please note that D_0 does not play a role for the value of t_2). The time scale t_2 for other temperatures T_2 and 3 different activation energies is shown in Fig. 1. Small time scales t_2 require high temperatures T_2 in the sediment and/or small activation energies.

$$t_2 = \frac{\exp\left(-\frac{E_a}{R T_1}\right)}{\exp\left(-\frac{E_a}{R T_2}\right)} t_1 \quad (3)$$

Figure 1: Time scale t_2 (Ma) for changes in ^{18}O isotopic composition as a function of the temperature experienced by CaCO_3 in sediments (Eq. 3).

The figure made by the Reviewer illustrates quite well why it is necessary to conduct experiments at temperatures higher than those encountered in natural settings. Eq. (2) reveals that the distance over which diffusion occurred during our experiments at 300 °C over 3 months would be reached after about 10 Ma to occur at a constant temperature of 25 °C. Unfortunately, translating diffusion lengths into magnitudes of re-equilibration requires a constant isotope disequilibrium to be maintained. This condition is not met in natural settings. In our experiments, foraminifera tests are immersed in pure ^{18}O water, the isotope disequilibrium is thus constant over time and does not depend on the temperature (which is also constant). In contrast, in natural settings, the isotope disequilibrium between foraminifera tests and pore water depends on the temperature (which varies over time). In fact, O isotope fractionation between calcite and water decreases with increasing temperature. It is this burial-induced increase of sediment temperature that establishes isotope disequilibrium between foraminifera tests and pore water. We thus cannot easily use these equations to simulate the isotope re-equilibration of foraminifera that occurs during sediment burial in natural settings.

Detailed comments/suggestions:

Abstract, L 25: 'seawater' is actually 'artificial seawater'

Done

L49 'warmer temperatures': temperatures can be low or high, however, not cold or warm (substances can be cold or warm)

Done

L72 Question: Do reported uncertainties refer to 1 or 2 σ ?

2 σ

L72-73 'artificial seawater': it would be good to give more information about the composition of the artificial seawater used in your experiments, especially with respect to concentrations of Ca^{2+} , Mg^{2+} , PO_4 , carbonate system (DIC, TA).

The 'artificial seawater' solution used for the experiments is deionized water that only contains 0.55 mol.L⁻¹ of NaCl and 0.003 mol.L⁻¹ of NaHCO₃. These details are provided as supplementary information.

L88 Question: 'pure Ca18O3' = Ca18O18O18O?

Yes

L113-114 'Because O isotope fractionation between calcite and water decreases with increasing temperature': here you could refer to Eq. 16.

Done

L149 time scale $t = 10^7$ years: How to derive this time scale? (compare discussion under 'Major comments/suggestions')

According to the present model, the $\delta^{18}\text{O}$ of foraminifera tests shifts by at least 0.1 ‰ or more (depending on the diffusion activation energy and the temperature gradient) during the first 10^7 years of burial. This value of 0.1 is significant compared to the typical uncertainty of bulk $\delta^{18}\text{O}$ measurements (0.05 ‰ [2σ]).

L165 'effect O' → 'effect of O'

Done

L181 'Yet, these processes has' → 'Yet, these processes have'

Done

L203-206 'At ambient temperature (293 K), the solution is slightly undersaturated with respect to calcite (within uncertainty). At this temperature, assuming thermodynamically controlled dissolution of calcite (i.e. neglecting kinetic barriers), equilibrium would be reached after dissolution of less than 2% of the tests.' Again: a bit more information about composition of the artificial seawater would be helpful.

The 'artificial seawater' solution used for the experiments is deionized water that only contains 0.55 mol.L⁻¹ of NaCl and 0.003 mol.L⁻¹ of NaHCO₃.

L233 'surface are' → 'surface area'

Done

Eq.(1) Can you please give a value (or range) for D_0 ?

The outputs of the present simulation do not depend on D_0 . Still, we can estimate a range for D_0 . Using the values reported by Anderson (1969) ($E_a = 71 \text{ kJ.mol}^{-1}$ and $D_0 = 4.6 \cdot 10^{-20} \text{ m}^2.\text{s}^{-1}$) in Eq. (15) yields $r_0 = 7 \text{ nm}$ (i.e. a grain size of about 14 nm assuming spherical grains), while a value of $r_0 = 8.3 \text{ }\mu\text{m}$ (i.e., grain size $\sim 16.5 \text{ }\mu\text{m}$) is obtained with the values reported by Farver and Yund (1998) ($E_a = 127 \text{ kJ.mol}^{-1}$ and $D_0 = 7.6 \cdot 10^{-9} \text{ m}^2.\text{s}^{-1}$).

Matching the actual grain size of foraminifera calcite domains (50 to 250 nm – Cuif et al., 2011) requires intermediate values of D_0 and E_a to be considered. This is easily explained by the large uncertainty (typically several tens of kJ.mol^{-1}) in the experimental determination of E_a values (Farver and Yund (1998) reported an uncertainty of $\pm 27 \text{ kJ.mol}^{-1}$ while Anderson (1969) did not report any) and by the large dispersion of foraminifera grain sizes and effective grain boundaries. A mixing coefficient x , comprised between 0 and 1, can be defined such as:

$$\begin{cases} E_{a,\text{true}} = xE_{a,\text{Farver}} + (1-x)E_{a,\text{Anderson}} \\ \log(D_{0,\text{true}}) = x\log(D_{0,\text{Farver}}) + (1-x)\log(D_{0,\text{Anderson}}) \end{cases}$$

Using these equations and Eq. (15), the foraminifera calcite grain size can be estimated as a function of x . As shown on Figure S1, x has to be comprised between 0.18 and 0.41 to yield grain size values from 50 to 250 nm, which corresponds to E_a values ranging from 81.4 to 94 kJ.mol^{-1} (i.e., the range used for the simulations) and D_0 values ranging from $4.8 \cdot 10^{-18}$ and $1.83 \cdot 10^{-15} \text{ m}^2.\text{s}^{-1}$.

Figure S1: Determination of the E_a and D_0 values for oxygen grain boundary diffusion in foraminifera calcites. (a) Calcite grain size values (r_0) obtained using Eq. (15) for different x values (Eq. (28)). (b) D_0 and E_a values obtained using Eq. (28) for different x values.

L250 'and T the temperature' → 'and T the absolute temperature'

Done

L255 'the equation 01' → 'Eq. (1)'

Done

L259 'where R is the ideal gas constant,' could be dropped (repetition)

Done

L259 'T_{xp} the temperature' → 'T_{xp} the absolute temperature'

Done

Initial and boundary conditions: the boundary condition for large x is missing (= 0 or flux = 0?)

C(x,t=0)=0, ∀x>0 (cf supplementary information).

Calling the molar ratio 18O/(18O +16O) a 'concentration' is a bit unusual. Why do you formulate the diffusion equation in terms of this ratio?

This formulation is classically used in O diffusion papers (e.g. Farver, 1994; Farver and Yund, 1998). The classical δ notation used by isotope geochemists would not help much in the present case because the water used for the present experiments only contained ¹⁸O, i.e. it had an "infinite" δ¹⁸O.

L273: 'Transport of 18O in foraminifera calcite follows Fick's 2nd law:' → 'The diffusion equation for C reads'. Fick's 2nd law: time change of concentration = divergence of flux

Done

Eq. (9): D_{foram} is a new notation that has not been introduced before

D_{foram} is introduced after Eq. (2) (cf Methods).

L311-312 'is equivalent to that of a step function with a length (d) of ...' → 'is equivalent to that of a step function of height C₀ with a length (d) of ...'

Done

L361-428 The 'spherical modelling section' (L361-428) needs a bit more explanation (what is the purpose of this modelling?) and could be shortened in terms of equations (discretisation of the diffusion eq. using finite differences is a standard procedure in numerical mathematics; you might cite a text book). This section essentially describes the discretisation of the diffusion equation in spherical coordinates and with spherical symmetry. It can be shortened quite a bit. I suggest to (1) give the appropriate equation, i.e.

$$\frac{\partial C}{\partial t} = \frac{1}{r^2} \frac{\partial}{\partial r} \left(r^2 \frac{\partial C}{\partial r} \right)$$

where r is the distance from the center of the sphere (the distance from the surface of the sphere is a rather unusual choice), (2) formulate the initial and boundary conditions, and (3) give the final discretisation formula.

Because the volumes impacted by the re-equilibration may not be negligible compared to the size of the foraminifera calcite building blocks, it was preferable to solve the diffusion equation with a spherical geometry rather than with a simpler 1D semi-infinite medium. We made this point clearer and simplified this section accordingly to the Reviewer's comment.

Note that the equation given by the Reviewer should read $\frac{\partial C}{\partial t} = \dots$ (and not $\frac{\partial C}{\partial r} = \dots$).

L371 'spherical corona' → 'spherical shell'

Done

Eq. (21) should read

$$V_i = \frac{4}{3}\pi \left[(r + dr/2)^3 - (r - dr/2)^3 \right]$$

Done

L382 'coronae' → 'shell'

Done

L465 I suggest to use a different notation for temperature given in ° C as, for example, T_C or θ .

Done (θ)

L476-481 'Again, although the bulk diffusion of oxygen in calcite is the sum of the contribution of both grain boundary and volume diffusion, it is commonly assumed that volume diffusion is slow and can be ignored^{35,36}. In fact, the activation energy for oxygen grain boundary diffusion in calcite aggregates was shown to be lower than that for oxygen volume diffusion in calcite single crystals (as low as 110-120 kJ.mol⁻¹ vs 175 kJ.mol⁻¹ – ref. 25).' The size of the diffusion coefficient depends on the activation energy, but also on the constant DO.

This is true (cf below)..

L489-490 'Intrinsic diffusion constant (D_0): Outputs of numerical simulations do not depend on the intrinsic diffusion coefficient.' I do not understand what is meant here. The dependency on the diffusion coefficient can vanish at steady state, however, I do not understand how that works in non-stationary states (look, for example, at Eq. (6)).

We demonstrate below that the diffusion length does not depend on D_0 in the case of the diffusion in a 1D semi infinite medium. Assuming that this length is small compared to calcite sizes, this demonstration can be expanded to the diffusion in spheres (see below):

If calcite crystals are assimilated to simple rods, the measured experimental ratio corresponds to:

$$Q = \frac{d}{r_0} = \frac{2}{r_0\sqrt{\pi}} \sqrt{D_{foram}t_{xp}}$$

Where r_0 is the length of the calcite domain. Combining this equation with Eq. (2) and (13) yields:

$$r_0 = \frac{2}{Q\sqrt{\pi}} \sqrt{D_0 \cdot \exp\left(-\frac{Ea}{RT_{xp}}\right)t_{xp}}$$

As emphasized in the manuscript, the integral of any diffusion profile can be assimilated to a step function with length d and height C_0 . Therefore, at any time, the instantaneous isotopic composition of a calcite ($\delta^{18}O_{cc}(t)$) can be expressed using the following mass balance:

$$\delta^{18}O_{cc}(t) = \frac{\delta^{18}O_{cc}(r=0,t) \cdot d + \delta^{18}O_{cc}(r,t=0) \cdot (r_0 - d)}{r_0},$$

which yields, after re-arrangement:

$$\delta^{18}\text{O}_{cc}(t) = \frac{d}{r_0} \left(\delta^{18}\text{O}_{cc}(0, t) - \delta^{18}\text{O}_{cc}(r, 0) \right) + \delta^{18}\text{O}_{cc}(r, 0)$$

Therefore, after a given duration t_{age} , the isotopic composition of the foram ($\delta^{18}\text{O}_{cc}^{overall}(t_{age})$) as defined in the ms. is:

$$\delta^{18}\text{O}_{cc}^{overall}(t_{age}) = \frac{1}{t_{age}} \frac{1}{r_0} \int_0^{t_{age}} d(\tau) \left(\delta^{18}\text{O}_{cc}(0, \tau) - \delta^{18}\text{O}_{cc}(r, 0) \right) d\tau + \delta^{18}\text{O}_{cc}(r, t = 0)$$

Replacing r_0 and $d(\tau)$ by their values taken from Eqs (2), (13) and the equation above yields:

$$\delta^{18}\text{O}_{cc}^{overall}(t_{age}) = \frac{1}{t_{age}} \frac{1}{\frac{2}{Q\sqrt{\pi}} \sqrt{D_0 \exp\left(-\frac{Ea}{RT_{xp}}\right)} t_{xp}} \int_0^{t_{age}} \frac{2}{\sqrt{\pi}} \sqrt{D_0 \exp\left(-\frac{Ea_{foram}}{RT_{xp}}\right)} \cdot \left(\delta^{18}\text{O}_{cc}(0, \tau) - \delta^{18}\text{O}_{cc}(r, 0) \right) d\tau + \delta^{18}\text{O}_{cc}(r, 0),$$

which yields, after re-arrangement:

$$\delta^{18}\text{O}_{cc}^{overall}(t_{age}) = \frac{1}{t_{age}} \frac{1}{\frac{1}{Q} \sqrt{\exp\left(-\frac{Ea}{RT_{xp}}\right)} t_{xp}} \int_0^{t_{age}} \sqrt{\exp\left(-\frac{Ea_{foram}}{RT(\tau)}\right)} \cdot \left(\delta^{18}\text{O}_{cc}(0, \tau) - \delta^{18}\text{O}_{cc}(r, 0) \right) d\tau + \delta^{18}\text{O}_{cc}(r, 0)$$

Note that this expression does not depend on D_0 .

In the case of spherical calcite, one can demonstrate that this result holds true, at least when the characteristic length of diffusion is negligible compared the calcite grain size (which eventually turned out to be the case for all of our simulations).

Let us re-write Eq. (13):

$$d = \frac{2}{\sqrt{\pi}} \sqrt{D_0 \cdot \exp\left(-\frac{Ea}{RT(t)}\right)} t = \sqrt{D_0} \cdot f(t)$$

As well as Eq. (15):

$$r_0 = \frac{2}{\sqrt{\pi}} \frac{\sqrt{D_0 \exp\left(-\frac{Ea_{foram}}{RT_{xp}}\right)} t_{xp}}{1 - (1 - Q)^{1/3}} = \sqrt{D_0} \cdot Cte_{xp}$$

where, for the sake of simplicity, all constant parameters of Eq. (15) but $\sqrt{D_0}$ were included in a term named Cte_{xp} .

If the characteristic length of diffusion is negligible compared the calcite grain size, one can calculate the instantaneous isotopic composition of calcite ($\delta^{18}\text{O}_{cc}(t)$):

$$\delta^{18}\text{O}_{cc}(t) = \frac{\delta^{18}\text{O}_{cc}(0,t)V_{shell} + \delta^{18}\text{O}_{cc}(r,0)V_{int}}{V_0} = \frac{\delta^{18}\text{O}_{cc}(0,t)4\pi(r_0-d)^2d + \delta^{18}\text{O}_{cc}(r,0)\frac{4\pi}{3}(r_0-d)^3}{\frac{4\pi}{3}r_0^3}$$

where V_{shell} stands a for a fully re-equilibrated external shell of calcite crystal, V_{int} is the internal volume unaffected by the re-equilibration, and V_0 is the initial volume. Replacing d and r_0 by their values taken from the expressions yields:

$$\delta^{18}\text{O}_{cc}(t) = \frac{\delta^{18}\text{O}_{cc}(0,t)4\pi(\sqrt{D_0}\cdot Cte_{xp} - \sqrt{D_0}\cdot f(t))^2\sqrt{D_0}\cdot f(t) + \delta^{18}\text{O}_{cc}(r,0)\frac{4\pi}{3}(\sqrt{D_0}\cdot Cte_{xp} - \sqrt{D_0}\cdot f(t))^3}{\frac{4\pi}{3}(\sqrt{D_0}\cdot Cte_{xp})^3}$$

which yields, after re-arrangement:

$$\delta^{18}\text{O}_{cc}^{overall}(t) = \frac{(\delta^{18}\text{O}_{cc}(0,t)(Cte_{xp} - f(t))^2f(t) + \frac{1}{3}\delta^{18}\text{O}_{cc}(r,0)(Cte_{xp} - f(t))^3)}{\frac{1}{3}Cte_{xp}^3}$$

As demonstrated here, $\delta^{18}\text{O}_{cc}^{overall}(t)$ does not depend on D_0 .

At the end, after a given duration t_{age} , the isotopic composition of a foraminifera ($\delta^{18}\text{O}_{cc}^{overall}(t_{age})$) is:

$$\delta^{18}\text{O}_{cc}^{overall}(t) = \frac{3}{t_{age}Cte_{xp}^3} \int_0^{t_{age}} \left(\delta^{18}\text{O}_{cc}(0,\tau)(Cte_{xp} - f(\tau))^2f(\tau) + \frac{1}{3}\delta^{18}\text{O}_{cc}(r,0)(Cte_{xp} - f(\tau))^3 \right) d\tau$$

We added this demonstration at the end of the section Methods of the revised version of our manuscript.

Reviewer #5

Review of "Burial-induced oxygen isotope re-equilibrium of fossil foraminifera explains ocean paleotemperature paradox."

The authors has succeed to make significantly important evaluations with new hypothesis on global trend of stable oxygen isotopic profile and there is no doubt that it will have a big impact on the audience. Because the $\delta^{18}\text{O}$ is quite basic tool for broad field of earth science, the influence of this study is not limited to paleoceanography, it is considered to be an important knowledge on the perspective of earth evolution for whole Cenozoic. This study give a clear answer to the hypothesis. The authors show a very good approach to solve this question with appropriate fancy methods of measurement and numerical modeling. The reviewer can identify the study is suitable for publishing in Nature Communication.

We thank the Reviewer for this positive comment and his/her support.

Major Questions

Have authors attempted oxygen isotopic mapping using foraminifera of the geological era (for example, from 40 Ma, 50 Ma)? If authors have already tried to visualize them, describe/indicate the result, regardless of success or failure. On the other hand, the reviewer imagines that such a measurement is not attempted. Perhaps the reviewer thinks that the effect of re-equilibrium could not be visualized by the current analytical accuracy of oxygen isotope ratio by NanoSIMS. Therefore, the reviewer speculates that it is necessary to conduct this simulation experiment of re-equilibrium in laboratory and computer. Even so, the author have to specify the reason why they does not show directly re-equilibrium imaging in the geological samples, in order to explain to the reader why the design of the experiment has become the current strategy.

Performing experiments using labelled water at 300°C was necessary to visualize isotope re-equilibration of foraminifera tests. In natural settings, given the relatively low temperatures undergone by foraminifera tests during sediment burial (~20-30 °C), the magnitude of isotope disequilibrium remains quite low (a few permil) and diffusion only occurs over small distances (a few nanometers). This cannot be visualized, even using high resolution technique such as NanoSIMS: re-equilibrated volumes are simply too small (NanoSIMS spatial resolution = 100 nm) and their isotope compositions are not enough different from the bulk to be precisely measured (NanoSIMS $\delta^{18}\text{O}$ precision = a few permils). We made this point clearer in the revised version of our manuscript (lines 61 to 65). Still, as illustrated by the present study, because of the small size of the calcite domains of foraminifera tests, diffusion can significantly impact their bulk O isotope composition over geologic timescales.

It is also necessary to explain about the reliability of the model has been guaranteed. The constructed diffusion model is the foundation in this study, but I could not find a description of how this correctness was evaluated. This is a just example, if authors enter the parameter of the present experiment in the numerical model, can they recursively check whether the model itself and the measured value are within same range or not, and whether reliable results can be obtained?

Using the present model to simulate the present experiments would require the algorithm to be extensively modified (the magnitude of the disequilibrium and the timescales are very different) and would thus not provide any guarantee regarding the reliability of the present model. Still, we did guarantee the reliability of the present model by recursively estimating each parameter while setting the others (i.e. the diffusion length, the foraminifera calcite size, the geothermal gradient and the activation energy).

Related to this, it is necessary to specify whether the crystal size obtained by equation 15 agrees with the known size by previous studies (50-200 nm, L119). This is also helpful in evaluating the correctness of the model. Though the authors has described "the mean calcite size can be calculated by solving for r_0 in the following equation.", but the reviewer can not find the calculation result.

Applied to the present experiments, Eq (15) yields grain sizes values comprised between 50 and 250 nm when using E_a values ranging from 81.4 to 94 $\text{kJ}\cdot\text{mol}^{-1}$ and D_0 values ranging from $4.8\cdot 10^{-18}$ and $1.83\cdot 10^{-15} \text{ m}^2\cdot\text{s}^{-1}$ (see above our answer to the comment of Reviewer #4 dealing with D_0). This is why we used these E_a values for the numerical simulations.

Does the same influence of re-equilibrium can be appeared in the carbon isotope ratio as well as the oxygen isotope ratio?

According to many studies, including the early ones of Anderson (1969, 1972), carbon diffusion in carbonates is quite similar to that of oxygen (Cherniak, Reviews in Mineralogy and Geochemistry, 2010). Thus, provided the presence of a second carbon partner in the system, foraminifera tests could undergo C isotope re-equilibration during burial. However, in the absence of direct evidence, we prefer to leave this question open.

L45: relatively cold tropical surface... I guess it is also newly explained in this study. Authors can mention about this wrong previous estimation in discussion as well as warmer estimation at higher latitude.

The Reviewer refers to the “cool tropics” paradox. Relatively cold tropical sea-surface temperatures were initially derived from the isotope composition of fossilized planktonic foraminifera tests. These estimations were recognized as biased by secondary calcite crystallization a few years ago. Still, the currently accepted interpretation of the planktonic foraminifera record is that the Paleogene equator-to-pole surface-ocean temperature gradient was much less steep than that of the present ocean. As stated in the introduction of our manuscript, this cannot be reconciled with the most recent climate and ocean circulation models and thus require explanation. Here, we show that during burial, diffusive isotope re-equilibration impacts the O isotope composition of fossil foraminifera tests that formed in cold waters (i.e. high-latitude planktonic species) more than tests formed in warmer water (i.e. tropical planktonic species). Corrected for burial-induced isotope re-equilibration, a steeper temperature gradient between low and high-latitude surface-ocean waters is re-established for the Paleogene, i.e. a gradient similar to the modern one.

L119: Iwasaki et al. (2015, Paleoclimatology: 10.1002/2014PA002639) also show clear image of calcite particles of planktonic foraminifera (*G. bulloides*). The reviewer thought this would be nice to see for authors.

We now refer to this study in the revised version of our manuscript.

L191: The species name would be Italicize.

Done

L566: The spelling will be fixed. 31. van der Lee, J. & De Windt, L. CHESS Tutorial and Cookbook. Updated for Version 3.0.

Done

Reviewer #4 (Remarks to the Author):

see attachment

Reviewer #5 (Remarks to the Author):

precisely read the revised manuscript "Burial-induced oxygen isotope re-equilibration of fossil foraminifera explains ocean paleotemperature paradoxes" by Bernard et al. All corrections are proper with my comments as well as others. Especially, the reviewer highly appreciate that the authors clearly indicate the reason why the combination of isotope labeling experiments and numerical experiments. The level of the manuscript is already high and this reviewer is convinced that this research will be the top science to publish on Nature Communications. It has been quite interesting for me to be able to participate in peer review on this manuscript.

Revision of “Burial-induced oxygen-isotope re-equilibration of fossil foraminifera explains ocean paleotemperature paradoxes” by Bernard, Daval, Ackerer, Pont, and Meibom.

Response to Referees

Reviewer #4

I am satisfied with the answer to my comments as well as with the revisions by the authors.

We thank the Reviewer #4 for his/her constructive comments that have been very helpful in improving our manuscript.

I still have a few comments (see below) that might help improving the manuscript. I recommend accepting the paper after minor revisions.

See below.

Change 'isotope composition' to 'isotopic composition'. The same applies for 'isotope fraction', 'isotope equilibrium', etcetera.

The Nature Research Editing Service which edited our manuscript for English language usage, grammar, spelling and punctuation approved the use of 'isotope composition'.

L138 'independently of the chemical composition of the pore water' I am not sure what you mean by this. My guess: you take into account temperature, but not changes in chemical composition of the pore water over time and depth, i.e. chemical composition of the pore water is considered constant?

Exact. We made this point clearer in the revised version of our manuscript.

The variable y is used in Eqs.(7), (8), and (10), however, has been introduced implicitly only. You could mention

$$y = \frac{x}{2\sqrt{D_0 \exp\left(-\frac{Ea_{foram}}{RT_{xp}}\right) t_{xp}}}$$

Done

P.S.: yes, the diffusion eq. should read

$$\frac{\partial C}{\partial t} = \frac{D}{r^2} \frac{\partial}{\partial r} \left(r^2 \frac{\partial C}{\partial r} \right)$$

We agree

Eq. (14) needs a bit more explanation. I guess it's a 'mass balance' for ^{18}O with on the left-hand-side of the equation a spherical shell (between $r_0 - d$ and r_0) consisting of ^{18}O , i.e. with $Q = 1$, and on the right-hand-side a sphere of radius r_0 with the observed mean Q .

Exact. We made this point clearer in the revised version of our manuscript.

L346: drop 'Fickian'

Done

L354: drop '(in _C)' (repetition)

Done

L364: drop '(in K)' (repetition)

We kept it, this is not a repetition.

L376-377 'where r represents the depth inside the spherical foraminifera calcite crystals (positive distance from the fluid/solid interface into the solid)' The radius r of spherical coordinates is usually defined as the distance from the origin of the coordinate system, here = centre of the spherical crystal. If you use a different definition (not recommended) you have to rewrite the diffusion equation (19). Later on (L 385) you refer to the usual definition of r : 'at a distance $r-dr$ from the centre'.

Exact. We corrected this point in the revised version of our manuscript.

L493 'assimilated to' is probably not the right term here; suggestions: replaced by, approximated by.

Done

L508 $r_0 \rightarrow r_0$

Done

Reviewer #5

Precisely read the revised manuscript "Burial-induced oxygen isotope re-equilibration of fossil foraminifera explains ocean paleotemperature paradoxes" by Bernard et al. All corrections are proper with my comments as well as others. Especially, the reviewer highly appreciate that the authors clearly indicate the reason why the combination of isotope labeling experiments and numerical experiments. The level of the manuscript is already high and this reviewer is convinced that this research will be the top science to publish on Nature Communications. It has been quite interesting for me to be able to participate in peer review on this manuscript.

We thank the Reviewer #5 for his/her constructive comments that have been very helpful in improving our manuscript.